# Mitigation of the double ITCZ syndrome in BCC-CSM2-MR through improving parameterizations of boundary-layer turbulence and shallow convection

Yixiong Lu[1], Tongwen Wu[1], Yubin Li[2], Ben Yang[3,4]

[1]Beijing Climate Center, China Meteorological Administration, Beijing, 100081, China
[2]School of Atmospheric Physics, Nanjing University of Information Science and Technology, Nanjing, 210044, China
[3]School of Atmospheric Sciences, Nanjing University, Nanjing, 210023, China
[4]CMA-NJU Joint Laboratory for Climate Prediction Studies, Nanjing University, Nanjing, 210023, China

*Correspondence to*: Yixiong Lu (luyx@cma.gov.cn)

**Abstract.** The spurious double intertropical convergence zone (ITCZ) is one of the most prominent systematic biases in coupled atmosphere-ocean general circulation models (CGCMs), and the underestimated marine stratus over eastern subtropical oceans has been recognized as a possible contributor. Rather than modifying the cloud scheme itself, this study significantly ameliorates the marine stratus simulation through improving parameterizations of boundary-layer turbulence and shallow convection in the medium-resolution Beijing Climate Center Climate System Model version 2 (BCC-CSM2-MR). The University of Washington moist turbulence scheme is implemented in BCC-CSM2-MR to better represent the stratocumulus, and a decoupling criterion is also introduced to the shallow convection scheme for improving the simulation of the stratocumulus-to-cumulus transition. Results show that the simulated precipitation in the eastern Pacific south of the equator is largely reduced, alleviating the double ITCZ problem. The tropical precipitation asymmetry index increases from −0.024 in the original BCC-CSM2-MR to 0.147 in the revised BCC-CSM2-MR, which is much closer to the observation. The study suggests that improving parameterizations of boundary-layer turbulence and shallow convection is effective for mitigating the double ITCZ syndrome in CGCMs.

## 1 Introduction

The coupled atmosphere-ocean general circulation models (CGCMs) have been widely used in the studies of climate variability, change and prediction. Despite decades of development, there are still many systematic biases in CGCMs, hampering the reliability of model results and limiting their utility. A prominent tropical bias in generations of CGCMs is the double intertropical convergence zone (ITCZ) syndrome, which is characterized by two parallel zonal bands of annual precipitation straddling the equator over the central and eastern Pacific, while it is absent in the observations (Mechoso et al., 1995; Lin, 2007; Zuidema et al., 2016; Zhang et al., 2019). Specifically, the double ITCZ bias is primarily seen in the Pacific and Atlantic sectors, and during the southern hemisphere rainy season (Li and Xie, 2014; Adam et al., 2018). The observed convergence

zone south of the equator extends southeastward from the western Pacific, whereas most CGCMs simulate a southern zonal rainfall band extending too far eastward. This bias is often associated with excessive warm sea surface temperatures (SSTs) in the southeastern Pacific (SEP). Similarly, in the tropical Atlantic basin, most CGCMs present a spurious southern zonal convergence zone, mirroring the actual zonal convergence zone north of the equator (Richter et al., 2014; Siongco et al., 2015).

The spurious double ITCZ not only affects the intensity of the Hadley circulation and the distribution of the trade winds directly related to the simulation of El Niño events, but also creates biases of latent heating in the tropics that can impact midlatitude weather and climate through atmospheric teleconnections (Schneider et al., 2009; Manganello and Huang, 2009).

The double ITCZ syndrome in CGCMs has been a long-standing problem, and it remains a serious impediment to model development. In earlier phases of Coupled Model Intercomparison Project (CMIP), from Phase 3 (CMIP3) to Phase 5 (CMIP5),
most CGCMs suffered from the double ITCZ problem to various degrees (Lin, 2007). Compared with CMIP3 models, there is no evidence of improvements in reducing the excessive precipitation and warmer SST in the SEP by the CMIP5 models, with results from CMIP5 somewhat worse than those from CMIP3 (Zhang et al., 2015). In the latest CMIP6, the biases persist in some of the current state-of-the-art CGCMs (Williams et al., 2018; Wu et al., 2019), indicating that it remains a tough challenge to alleviate the double ITCZ bias in CGCMs.

Many efforts have been devoted to identifying the possible contributors to the double ITCZ problem, and both oceanic and atmospheric causes are suggested (Zhang et al., 2019). For instance, by restoring model ocean temperature and salinity to observations in the upper ocean, the effects of ocean eastern boundary biases along the Peruvian-Chilean coast was investigated, and it was found that the coastal SST and salinity biases exert significant influences on the SEP precipitation (Large and Danabasoglu, 2006). Modelling studies show that the double ITCZ bias is very sensitive to atmospheric processes, such as
convection and cloud radiative effects. As ITCZ precipitation originates from convections, convection parameterization schemes are often blamed for the double ITCZ problem. By modifications in closure, trigger function, and lateral entrainment of the convection scheme, the double ITCZ can be mitigated to varying degrees (Song and Zhang, 2009; Zhang and Song, 2010; Queslati and Bellon, 2013; Song and Zhang, 2018). Deficiencies in the extratropical cloud simulation are also suggested to be possible causes of the tropical double ITCZ (Hwang and Frierson, 2013; Li and Xie, 2014). It is argued that the negative
cloud amount biases over the Southern Ocean and the associated warming can induce anomalous northward cross-equatorial atmospheric energy transport, resulting in a meridional shift of ITCZ. However, based on this atmospheric teleconnection argument and by reducing the shortwave radiation bias over the Southern Ocean in the fully coupled National Center for Atmospheric Research (NCAR) Community Earth System Model version 1 (CESM1), Kay et al. (2016) found that the double ITCZ is not improved.

The negative cloud amount biases off the west coast of South America, which are common in CGCMs, are regarded as another cause for the double ITCZ problem (Dai et al., 2003, 2005). It is believed that the underestimated cloud cover leads to more net heat flux into the ocean and the warm SST biases in the SEP, which is associated with stronger convections and precipitation. Previous attempts to ameliorate the double ITCZ bias mostly focused on the role of the cloud fraction parameterization, either through prescribed increases or through physical parameterization changes (Ma et al., 1996; Yu and

Mechoso, 1999; Dai et al., 2003, 2005; Qin and Lin, 2018). Furthermore, increased cloud fraction and the associated shortwave cloud forcing over the SEP can be driven by changes in representation for cloud microphysics (Woelfle et al., 2019). Modifications on the cloud scheme itself are helpful to mitigate the double ITCZ to some extent; however, it still plagues most CGCMs.

Rather than directly modifying the cloud macro/microphysics schemes in CGCMs, this study changes parameterizations
of the boundary-layer turbulence and shallow convection to improve the low-level cloud simulation indirectly. The low-level cloud near the South American west coast is the steadiest and most persistent stratocumulus regime in the world (Klein and Hartmann, 1993; Wood and Bretherton, 2006), which is closely related to multiple processes, such as boundary layer (BL) mixing, surface sensible and latent heat fluxes, cloud-top radiative cooling, and entrainment (Wood 2012). Furthermore, a prominent feature of the low-level cloud in the SEP is that the stratocumulus regime progressively transforms into the trade
cumulus regime moving downstream off the coast. In the transition, interactions between BL turbulence and shallow convection play a critical role. When multiple separate BL and shallow convection schemes are combined to complement each other in a CGCM, inconsistencies between parameterizations are likely to be introduced. This study aims to find out whether improving parameterizations of BL turbulence and shallow convection alleviates the double ITCZ bias in CGCMs. This is done through a two-step process, using the medium resolution Beijing Climate Center Climate System Model version 2 (BCC-
CSM2-MR). The first step is to determine if the stratocumulus simulation is improved with prescribed SSTs through modifying parameterizations of BL turbulence and shallow convection in the atmospheric component of BCC-CSM2-MR. The second step, which indicates the atmosphere-ocean feedback, is to perform a pair of ocean-atmosphere coupled simulations to demonstrate the impact of improved stratocumulus representation on the simulation of the SSTs and precipitation in the SEP.

The paper is organized as follows. Section 2 describes the BCC-CSM2-MR, focusing on the new configurations of the
BL and shallow convection schemes, as well as the model setups for the control and sensitive experiments. A brief review of the observational data used to evaluate the model results is also included. Section 3 compares the cloud simulation results from the atmosphere-only runs, and section 4 discusses the impact of improved cloud simulation on the SSTs and precipitation in the coupled runs. Further discussion on the effectiveness of the new configuration of the BL turbulence and shallow convection parameterizations is shown in section 5. At last, section 6 presents the summary and conclusions.

**2 Model description, experimental design, and observational data**

**2.1 Brief description of the BCC-CSM2-MR**

In the present study, we use the version of BCC-CSM2-MR participating in CMIP6, which is a fully coupled model with atmosphere, ocean, land surface, and sea ice components (Wu et al., 2019). The atmospheric component in BCC-CSM2-MR is based on the Beijing Climate Center atmospheric general circulation model (BCC-AGCM; Wu et al., 2010). BCC-AGCM
originates from the community atmospheric model version 3 (CAM3) developed by the National Center for Atmospheric Research (NCAR), but has evolved into a largely different model. The spectral dynamical core of BCC-AGCM is featured by

introducing a reference stratified atmospheric temperature and a reference surface pressure into the governing equations (Wu et al., 2008). Besides, model physics in BCC-AGCM has been substantially updated, including a new deep convection scheme (Wu, 2012), modified parameterizations for cloud cover, a revised algorithm for the air-sea turbulent fluxes, an empirical equation to compute the snow cover fraction, etc. The vertical discretization of BCC-AGCM also differs from CAM3 (Wu et al., 2019). The Beijing Climate Center Atmosphere-Vegetation Interaction Model (BCC-AVIM; Li et al., 2019) serves as the land component of BCC-CSM2-MR. It includes major land surface biophysical and plant physiological processes. The oceanic component is based on the Modular Ocean Model version 4 (MOM4; Griffies et al., 2005) and the sea ice component is the Sea Ice Simulator (SIS; Winton, 2000). Over the sea ice, a new bulk aerodynamic algorithm is formulated for computing surface exchange fluxes (Lu et al., 2013). The above four components are physically coupled through fluxes of momentum, energy, and water at their interfaces, which is realized using the NCAR flux coupler version 5.

The atmospheric component of BCC-CSM2-MR has a horizontal resolution of T106 (approximately 1.125° latitude by 1.125° longitude) with 46 hybrid vertical levels. The deep convection scheme used in BCC-CSM2-MR is based on a mass-flux bulk cloud model approach, in which the mass change for the adiabatic ascent cloud parcel with altitude is derived from a total energy conservation equation of the whole adiabatic system involving the updraft cloud parcel and the environment (Wu, 2012). Compared with its previous version in CMIP5, BCC-CSM2-MR has notably improved the simulation skills of atmospheric variability in the tropics, such as the Madden-Julian oscillation and the stratospheric quasi-biennial oscillation (Wu et al., 2019; Lu et al., 2020).

However, the version of BCC-CSM2-MR participating in CMIP6 still suffers from the double ITCZ syndrome. The mean precipitation errors are dominated by systematic errors along the ITCZ. Here we show that a revised version of BCC-CSM2-MR, configured with new parameterizations of BL turbulence and shallow convection, significantly reduces the double ITCZ bias.

## 2.2 Parameterization of BL processes

The CMIP6 configuration of the BCC-CSM2-MR employs the Holtslag and Boville (1993) parameterization (hereafter called the HB scheme), which is optimized for the simulation of dry convective BLs over land. The HB scheme is based on the eddy diffusivity approach. And the eddy diffusivity of variables $\chi$ is given by

$$K_{\chi} = k w_{\chi} z \left(1 - \frac{z}{h}\right)^2 , \qquad (1)$$

where $k$ is the von Karman constant; $z$ is the height; $w_{\chi}$ is a turbulent velocity; and $h$ is the boundary layer height. For neutral and stable conditions, $w_{\chi}$ is proportional to the friction velocity, while for unstable conditions, $w_{\chi}$ is proportional to the convective velocity scale $w_*$

$$w_* = \left[\frac{g}{\theta_{vs}} h \overline{\left(w'\theta_v'\right)_s}\right] . \qquad (2)$$

Here, $g$ is the gravitational acceleration; $\theta_{vs}$ is the virtual potential temperature at surface; and $\overline{\left(w'\theta_v'\right)}_s$ represents the buoyancy heat flux at surface. The above formula suggests that the HB scheme assumes the BL turbulence to be forced exclusively from the surface heating and friction velocity.

In the marine stratocumulus-capped BLs, the turbulence structure depends strongly on the dominant turbulence generating mechanism resulting from both evaporative and radiative cooling at cloud top. To provide a more physically realistic treatment of stratocumulus-topped BLs, the University of Washington moist turbulence (UWMT) scheme from Bretherton and Park (2009) is implemented in the revised BCC-CSM2-MR to replace the HB scheme. The UWMT scheme also uses first-order $K$ diffusion to represent all the turbulence, in which the eddy diffusivity is calculated based on the turbulent kinetic energy (TKE, $e$) and proportional to the stability-corrected length scale $lS_\chi$, given by

$$K_\chi = lS_\chi \sqrt{e} .$$
(3)

In the case of an inversion layer at the top of convective BLs, the diffusivities are parameterized as following

$$K_\chi = w_e \Delta z_e ,$$
(4)

where $w_e$ is the entrainment rate, and $\Delta z_e$ is the thickness of the entrainment layer. The UWMT scheme uses the $w_*$ entrainment closure raised by Nicholls and Turton (1986):

$$w_e = A \frac{w_*^3}{\left(g\Delta^E s_{vl}/s_{vl}\right)\left(z_t - z_b\right)} .$$
(5)

Here, $z_t$ and $z_b$ are the top and bottom heights of the entrainment layer, respectively; $\Delta^E$ denotes a jump across the entrainment layer; and $s_{vl}$ is the liquid virtual static energy. $A$ is a nondimensional entrainment efficiency, which is affected by evaporative cooling of mixtures of cloud-top and above-inversion air. Following Bretherton and Park (2009), $A$ is expressed as

$$A = 0.1\left(1+30E\right) ,$$
(6)

where $E$ is the evaporative enhancement, which is parameterized as

$$E = 0.8Lq_l^{ct}/\Delta s_{vl} .$$
(7)

$L$ is the latent heat of vaporization, $q_l^{ct}$ is the cloud-top liquid water content, and $\Delta s_{vl}$ is the jump in the liquid virtual static energy across the cloud-top entrainment zone.

**2.3 Shallow cumulus Parameterization**

To treat shallow cumulus, the original version of the BCC-CSM2-MR adopts a stability-dependent mass-flux representation of moist convective processes with the use of a simple bulk three-level cloud model, as in Hack (1994). Specifically, in a vertically discrete model atmosphere where the level index $k$ decreases upward and layers $k$ and $k+1$ are moist adiabatically

unstable, the Hack scheme assumes that there is a non-entraining convective element originated from level $k+1$, condensation and rain out processes in level $k$, and limited detrainment in level $k-1$. By repeated application of this procedure from the bottom of the model to the top, the thermodynamic structure is locally stabilized.

The Hack shallow cumulus scheme can be also active in moist turbulent mixing, such as stratocumulus entrainment, which has different physical characteristics than cumulus convection. Shallow cumulus is usually regarded as a decoupled BL regime, in which the vertical mixing processes do not achieve a single well-mixed layer, while the stratocumulus regime represents a well-mixed BL up to the cloud top. The decoupling criterion to distinguish the two regimes is of great importance for simulating the stratocumulus-to-cumulus transition (Bretherton and Wyant, 1997; Wood and Bretherton, 2004). A number of these decoupling criteria have been explored, such as static stability (Klein and Hartmann, 1993) and the buoyancy flux integral ratio (Turton and Nicholls, 1987). In the light of its robustness, the stability criterion with a threshold of 15 K is introduced into the Hack scheme. The lower tropospheric stability (LTS) is defined as

$$LTS = \theta_{700hPa} - \theta_{sfc} \ , \tag{8}$$

where $\theta_{700hPa}$ and $\theta_{sfc}$ are potential temperatures at 700 hPa and surface, respectively. In the revised BCC-CSM2-MR, the Hack scheme is activated only in the decoupled BL regimes with $LTS < 15$ K below 700 hPa. Above 700 hPa, the Hack scheme is retained to remove any local instability as long as the two adjacent model layers are moist adiabatically unstable. It should be noted that the LTS criterion has been developed into physically more plausible formula. Wood and Bretherton (2006) modified the LTS to account for the strength of the BL inversion, called the estimated inversion strength (EIS) which is shown to be more useful than LTS for determining low cloud cover in the present climate. EIS is then further revised to take into account cloud-top entrainment and transformed into the estimated cloud-top entrainment index (ECTEI), which shows dependence on sea surface temperature (Kawai et al., 2017). Impacts of more sophisticated criteria on cloud representation and precipitation simulation in BCC-CSM2-MR is beyond the scope of this paper and will be explored in future work.

## 2.4 Cloud fraction parameterization

In the original BCC-CSM2-MR, the cloud amount is evaluated via a diagnostic method, depending on relative humidity, convective mass fluxes and atmospheric stability (Wu et al., 2019). Three types of clouds are diagnosed: layered stratus cloud, convective cloud, and low-level marine stratocumulus. The marine stratocumulus cloud fraction is particularly parameterized to compensate the HB dry turbulence scheme. It is calculated by using an empirical relationship between the observed stratocumulus cloud amount and the boundary layer stratification, which is evaluated with potential temperatures at 700 hPa and surface, as in Klein and Hartmann (1993).

Considering the advantage of the UWMT scheme to simulate marine stratocumulus, the revised BCC-CSM2-MR excludes the empirical calculation of stratocumulus cloud amount and only uses the relative humidity and convective cloud

fraction to deduce the overall cloud amount. The stratocumulus cloud fraction is assumed to be diagnosed from relative humidities. The critical relative humidities $RH_c$ are used to tune the global annual mean top-of-atmosphere (TOA) shortwave and longwave radiative energy fluxes close to observations. The original BCC-CSM2-MR uses the low-level (below 750 hPa) $RH_c$ of 0.945, while the revised BCC-CSM2-MR uses the low-level $RH_c$ of 0.97. Given the exclusion of special stratocumulus cloud fraction parameterization and increment in the low-level $RH_c$ in the revised BCC-CSM2-MR, better simulation skills of

low-level cloud are attributed to the improved schemes of BL turbulence and shallow convection.

## 2.5 Experimental design

Table 1 summarizes all the experimental setups. Two sets of 11-yr Atmospheric Model Intercomparison Project (AMIP)-type simulations are conducted with the same prescribed sea surface boundary conditions using the Hadley Centre SSTs and sea ice concentrations. The one using the default configurations of HB and Hack schemes is referred to as REF_amip, and the

195 other using the new configurations of UWMT and modified Hack schemes is referred to as NEW_amip. For each integration, the first year is treated as the spinup, and the last 10 years are used for analysis. Both runs start from identical initial conditions. Note that the Cloud Feedback Model Intercomparison Project Observation Simulator Package (COSP; Bodas-Salcedo et al., 2011) is turned on to better compare with satellite observations.

For the fully coupled experiments, two sets of 12-yr Coupled Model Intercomparison Project (CMIP)-type simulations

are carried out under pre-industry conditions, using greenhouse gases, ozone, aerosol emission, and others fixed at the level of 1850. The CMIP simulation with the default HB and Hack schemes is referred to as REF_cmip, whereas the simulation using the UWMT and modified Hack schemes is referred to as NEW_cmip. Both runs are initialized with the data sets at the 345th year of a pre-industry control run provided by the default BCC-CSM2-MR. Theses initialization data sets are close to the equilibrium state. Note that previous studies have shown that the formation and mitigation of the double ITCZ biases can be

completed within the first 2 years of simulation (Liu et al., 2012). Therefore, a 12-yr integration for each coupled experimental setup is performed, of which the first 2 years are treated as the spinup, and the last 10 years are used for analysis.

To highlight the relative roles of the UWMT scheme and the modified Hack scheme in the low-level cloud simulation in NEW_amip, two extra AMIP-type sensitivity experiments are designed. The modifications of the boundary-layer turbulence scheme and the shallow convection scheme are added, respectively in the two sensitivity experiments. The simulation with

210 changes only in the boundary-layer turbulence scheme is referred to as UWMT_amip, whereas the simulation with changes only in the shallow convection scheme is referred to as mHack_amip. The difference between UWMT_amip and REF_amip will show the isolate impact of change in the boundary-layer turbulence parameterization, while the difference between mHack_amip and REF_amip will show the isolate impact of change in the shallow convection parameterization.

## 2.6 Observational data

To evaluate the simulated cloud amount and shortwave cloud radiative forcing (SWCRF), the distribution of cloud fraction from the CALIPSO GOCCP data set (GCM-Oriented CALIPSO Cloud Product; Chepfer et al., 2010) and TOA SWCRF from

the CERES-EBAF data set (Clouds and the Earth's Radiant Energy System Energy Balanced and Filled data product; Loeb et al., 2018) are used as references. The simulated precipitation is compared with the Global Precipitation Climatology Project (GPCP) monthly precipitation analysis (Adler et al., 2003) for years 1981 to 2010. Considering the uncertainty in precipitation observations, the Climate Prediction Center (CPC) Merged Analysis of Precipitation (CMAP; Xie and Arkin, 1997) from 1981 to 2010 is also used. Additional observations include SSTs from the Hadley Centre Sea Ice and Sea Surface Temperature data set (HadISST; Rayner, 2003), and wind velocities from the Japanese Meteorological Agency 55-year Reanalysis (JRA-55; Kobayashi et al., 2015).

## 3 Improved cloud representation in the atmosphere-only simulations

### 3.1 TOA SWCRF

TOA SWCRF is one of the most important metrics to examine for improving the representation of low-level clouds. Figure 1 compares annual mean TOA SWCRF simulated by REF_amip with CERES-EBAF observations. The global mean SWCRF is −45.83 W m$^{-2}$ in CERES-EBAF and −49.51 W m$^{-2}$ in REF_amip. REF_amip simulates the SWCRF distributions relatively well, but overestimates the magnitude of SWCRF over regions of Northern Hemisphere storm tracks and underestimates the magnitude of SWCRF over the subtropical marine stratocumulus regions. With new configurations of BL turbulence and shallow convection schemes, NEW_amip presents a global mean TOA SWCRF of −54.45 W m$^{-2}$ (Figure 1c). Compared with REF_amip (Figure 1f), NEW_amip shows a considerably increased magnitude of SWCRF over the eastern subtropical ocean regions, suggesting that the representation of low-level marine stratocumulus is enhanced. However, the SWCRF is overestimated in NEW_amip over these regions (Figure 1e).

### 3.2 Low-level cloud cover

TOA SWCRF strongly depends on the low-level cloud fraction. As shown in Figure 2a, the observed low cloud is prevalent over midlatitude storm-track regions, eastern tropical and subtropical oceans and the Southern Ocean. Figure 2b and 2c present the simulated annual mean low-level cloud amount from REF_amip and NEW_amip, respectively. Generally, both simulations reproduce the observed patterns of low-cloud amount with maxima in the midlatitude storm tracks and minima in the trade cumulus regime over the ocean. One prominent discrepancy between REF_amip and observations is a remarkable underestimation of low-cloud fraction over the eastern subtropical oceans, which contributes to the weak bias in the magnitude of TOA SWCRF over these regions in REF_amip (Figure 1d). The observed maxima in the subtropical stratocumulus decks are better simulated by NEW_amip. The global mean low-level cloud fraction increases from 24.64% in REF_amip to 28.74% in NEW_amip, closer to 37.06% in the GOCCP observations. With new configurations of BL turbulence and shallow convection parameterizations, the low-cloud fraction is significantly increased by up to 40% over the eastern subtropical oceans (Figure 2d). As shown in Burls et al. (2017), increased cloud fraction in the subtropical eastern Pacific has an important effect on the cold tongue by cooling sea surface waters which subduct and eventually end up in the equatorial Pacific.

For more quantitative comparisons, Table 2 presents the area-averaged biases and root-mean-square errors (RMSEs) of the REF_amip and NEW_amip low cloud simulations to the GOCCP observations over the globe, in the tropics and for the five main subtropical marine stratocumulus regions shown in Figure 2. For all regions, REF_amip significantly underestimates the low cloud amounts and has large biases and RMSEs. Although the low cloud cover simulated by NEW_amip is still less, biases and RMSEs are substantially reduced for most regions, except for Canara where the cloud fraction is overestimated to some extent. Spatial pattern correlations are also calculated to evaluate the simulated low cloud distribution. For the global low cloud pattern, the correlation increases from 0.76 in REF_amip to 0.84 in NEW_amip. More obviously, the tropical pattern correlation increases from 0.72 in REF_amip to 0.89 in NEW_amip. Based on these objective measures, it is clear that NEW_amip performs better than REF_amip with improved parameterizations of BL turbulence and shallow convection.

### 3.3 The subtropical Stratocumulus-to-Cumulus (Sc-to-Cu) transition

In order to further illustrate the consistency between the UWMT turbulence scheme and modified Hack shallow convection scheme, Figure 3 shows vertical cross sections of cloud fraction depicting the subtropical stratocumulus to trade cumulus transition both in the observations and simulations. We focus on the five main regions of the subtropical Sc-to-Cu transition, with the locations shown in Figure 2a. Observational guidance is provided from GOCCP. GOCCP generally shows a clear transition from stratocumulus near the coast to trade cumulus well offshore, marked by a gradual reduction of cloud cover along with a rising cloud-top height and thickening cloud depth moving downstream off the coasts (Figure 3, right). However, REF_amip (Figure 3, left) tends to overall underestimate the cloud amount through the cross sections. Specifically, REF_amip produces the cloud fraction much less near the coast and fails to simulate the elevated maxima prevalent in the observed cross sections. This reflects the limitations of the HB turbulence scheme in simulating stratocumulus. In addition, REF_amip is prone to abruptly reduce the cloud amount out of the coastal stratocumulus regions, resulting in an almost cloudless state offshore. No clear transitional regime is simulated, which can be attributed to the excessive vertical mixing from the original Hack shallow convection scheme over the western edge of the stratocumulus regime. In contrast to REF_amip, the cross sections in NEW_amip depict a very different picture. With new configurations of boundary-layer turbulence and shallow cumulus schemes, NEW_amip presents more realistic characteristics of subtropical Sc-to-Cu transition (Figure 3, middle). The qualitative aspects of the transition are better captured for all regions: most notably the gradual reduction of cloud cover as it moves offshore and the placement of the maximum cloudiness which generally occurs somewhat offshore, as in GOCCP. A possible reason for the over-extension of Sc in NEW_amip may be that the decoupling criterion added to the Hack shallow convection scheme is too strong, leading to the weak vertical mixing over the eastern edge of the shallow cumulus regime. The weak vertical mixing may further suppress the upward transport of water vapour, resulting in excessive low-level cloud amounts.

**4 Alleviated double ITCZ in the coupled simulations**

The underestimated stratus cloud cover in the SEP has long been recognized as a possible contributor to the double ITCZ bias in the eastern Pacific. The above-mentioned analysis from the atmosphere-only runs demonstrates that the simulated low-cloud amount in the SEP can be increased obviously by modifying parameterizations of BL turbulence and shallow convection. Is it possible that the improved cloud simulation reduces the warm bias of SST in the SEP and hence helps to eliminate the double ITCZ bias? Below we examine the effects of the increased low-cloud amount in the coupled simulations.

**4.1 Precipitation**

**4.1.1 Precipitation patterns**

Figure 4 shows simulated annual-mean precipitation distributions compared with the GPCP and CMAP observational estimates. Two sets of observational estimates are presented here to illustrate the uncertainty in observations. In observations (Figure 4a and 4b), massive precipitation can be found in regions of Asian monsoon and midlatitude storm tracks over the northwest Pacific and Atlantic Oceans. In the tropics, the primary peaks are located in the eastern Indian Ocean and Maritime Continent regions. Furthermore, two zonal precipitation bands are located at $0° – 10°N$ in the equatorial Pacific and Atlantic oceans, respectively, constituting the northern ITCZ. The southern South Pacific convergence zone (SPCZ) is mainly located around $5°S – 10°S$ near the western Pacific warm pool region and experiences a southeast tilt as it extends eastward into the central Pacific. In the southeast Pacific, the SPCZ, the west coast of South America, and the northern ITCZ bound a triangular-shaped dry region, which is dominated by stratocumulus and trade cumulus. While the main spatial patterns of observed precipitation climatology are properly reproduced, prominent double ITCZ biases develop in the simulation of tropical precipitation in REF_cmip (Figure 4c). Specifically, the simulated SPCZ does not tilt southeastward as that in the observation but overly extends eastward, leading to the excessive precipitation in the central and eastern equatorial Pacific. The precipitation band of $> 3$ mm day$^{-1}$ between $10°S$ and $5°S$ extends as far east as $90°W$. Also, a spurious precipitation band appears in the southern Atlantic at $10°S – 0°$. In contrast, NEW_cmip significantly reduces the SPCZ bias in the central and eastern equatorial Pacific and produces no rain belt in the tropical southern Atlantic (Figure 4d). The precipitation of $> 3$ mm day$^{-1}$ between $10°S$ and $5°S$ is confined to west of $130°W$ in the Pacific Ocean. A more detailed plot with differences of annual-mean precipitation rates between NEW_cmip and REF_cmip is shown in Figure 5. The differences are mainly presented in the tropics, with decreased precipitation in the western Pacific, the SPCZ, the equatorial Atlantic oceans over $0−10°S$, and western Indian ocean. Particularly, the precipitation rates over the SPCZ and the equatorial southern Atlantic are reduced up to 4 mm day$^{-1}$, leading to an alleviation of the double ITCZ bias in the NEW_cmip experiment.

Table 3 summarizes the area-averaged biases and RMSEs, and pattern correlations between simulated and observed precipitation rate in the tropical Pacific. Compared with GPCP (CMAP), the bias of simulated precipitation rate is reduced from 0.89 (0.33) in REF_cmip to 0.44 (-0.12) in NEW_cmip. Correspondingly, the RMSE decreases from 0.94 (0.48) in REF_cmip to 0.54 (0.36) in NEW_cmip. The elimination of excessive precipitation in the SEP leads to an increase of the

pattern correlation, which is raised from 0.78 (0.80) in REF_cmip to 0.81 (0.81) in NEW_cmip. It is also interesting to note that the spurious southern precipitation belt in the equatorial Atlantic completely disappears in NEW_cmip, which agrees well with observations.

It should be noted that precipitation simulation is a complex problem, involving many processes such as deep convection and cloud microphysics. The modification of boundary layer and shallow convection schemes in the model will affect the

performance of deep convection and cloud microphysical schemes, and then cause changes in precipitation simulation. For example, a negative bias in the equatorial Indian Ocean seems to get worse in NEW_cmip, which may be due to the indirect effects of changes in boundary layer and shallow convection parameterizations. Also, it is interesting and worth highlighting that the cold tongue bias, which is closely linked to the double ITCZ bias, persists in NEW_cmip, implying that other parameterized processes, e.g., deep convection and oceanic circulations, may play an important role in achieving more

improvements.

### 4.1.2 Asymmetry of the tropical precipitation

The asymmetry in the meridional distribution of tropical precipitation is a characteristic in the observed ITCZ. Figure 6 shows the zonally averaged annual mean precipitation from the GPCP and CMAP observations, and the REF_cmip and NEW_cmip simulations. The CMAP observations generally show very similar latitudinal variations to that of GPCP, but with slightly

stronger precipitation intensity. These two sets of observational estimates both show a pronounced asymmetry about the equator with strong northern ITCZ peak at 5° – 10°N and weak southern ITCZ peak at 10° – 5°S. The precipitation produced by REF_cmip is about 50% larger than the observed amounts in the southern ITCZ, which comes largely from the spurious rainfall in the central and eastern Pacific and Atlantic south of the equator. The precipitation rate in the southern ITCZ is comparable to that in the northern ITCZ, and the meridional distribution of the tropical precipitation tends to be symmetrical.

In contrast, NEW_cmip produces much weaker precipitation in the southern ITCZ, which is in good agreement with the observed asymmetry.

To quantitatively examine the ITCZ fidelity in the coupled simulations, the tropical precipitation asymmetry index ($A_P$), which is defined as the precipitation difference between the northern (0° – 20°N) and southern (20°S – 0°) tropics normalized by the tropical mean, is calculated as follows (Hwang and Frierson, 2003; Xiang et al., 2017; Adam et al., 2018)

$$A_P = \left( \overline{P}_{0\text{–}20°\text{N}} - \overline{P}_{20°\text{S}\text{–}0} \right) \Big/ \overline{P}_{20°\text{S}\text{–}20°\text{N}} \, , \tag{9}$$

where the overbar represents a zonal mean of precipitation. The observed annual mean $A_P$ is 0.194 in GPCP and 0.214 in CMAP, respectively, because the ITCZ is predominantly north of the equator. In the coupled simulations, the $A_P$ index increases from −0.024 in REF_cmip to 0.147 in NEW_cmip, which is much closer to the observed values, primarily because the southern ITCZ in the Pacific and Atlantic sectors is reduced.

On the other hand, the symmetric component of the tropical precipitation is quantified using the equatorial precipitation index $E_p$, defined as (Adam et al., 2016, 2018)

$$E_P = \frac{\overline{P}_{2^\circ S - 2^\circ N}}{\overline{P}_{20^\circ S - 20^\circ N}} - 1 . \tag{10}$$

In the case of double ITCZ that straddle the equator and when the equatorial precipitation vanishes, $E_p$ assumes its minimum value $E_p = -1$. The more strongly peaked tropical precipitation is on the equator, the larger $E_p$. $E_p$ is also found to be largely correlated with the difference in zonal mean precipitation between the absolute maximum and the equator (Popp and Lutsko, 2017). The observed equatorial precipitation indices are 0.136 in GPCP and 0.110 in CMAP, respectively, whereas the simulated values are much smaller, which is 0.013 in REF_cmip and further reduces to −0.008 in NEW_cmip. The worse index in NEW_cmip is consistent with less equatorial precipitation shown in Figure 5.

### 4.1.3 Seasonal cycle of precipitation over eastern Pacific

Figure 7 presents the seasonal evolution of monthly precipitation averaged between 90°W and 160°W from the GPCP observations and coupled simulations. In the observation (Figure 7a), the precipitation greater than 3 mm day$^{-1}$ occurs only between March and April south of the equator, whereas the precipitation greater than 3 mm day$^{-1}$ persists throughout the year in the northern ITCZ region with higher precipitation rates occurring between April and November. The double ITCZ bias in REF_cmip comes mainly from overestimated precipitation as high as 12 mm day$^{-1}$ between 5°S and 10°S in boreal winter and spring (Figure 7b). In contrast, the excessive precipitation south of the equator in boreal winter and spring is reduced in NEW_cmip and closer to the observation although the maximum rainfall is slightly larger in April (Figure 7c and 7d), indicating the alleviated double ITCZ bias. In addition, the northern ITCZ dry bias in the boreal winter and spring is also alleviated and the northern precipitation band in boreal summer and autumn moves closer to the equator in NEW_cmip.

A southern ITCZ (*SI*) index, simply defined as the annual mean precipitation rate over southeastern Pacific (20°S – 0°, 90° – 160°W; Bellucci et al., 2010), is used to quantify the coupled model biases in a more objective way. The simulated *SI* index decreases from 3.46 mm day$^{-1}$ in REF_cmip to 2.51 mm day$^{-1}$ in NEW_cmip, closer to the observational values (1.36 mm day$^{-1}$ in GPCP and 1.66 mm day$^{-1}$ in CMAP). The following analysis will focus on the Pacific ocean to understand the impact of improved boundary-layer turbulence and shallow convection schemes on the simulated ITCZ.

### 4.2 SST and the heat flux into the ocean

The annual mean precipitation change is closely related to the SST change (Song and Zhang, 2016). Figure 8 shows the annual mean SST from the HadISST observation, the REF_cmip and NEW_cmip simulations, and their differences in the Pacific. The 30-year average of HadISST from 1971 to 2000 is used as the observed climatology, in which the warm SST corresponds to the northern ITCZ and SPCZ precipitation (Figure 8a). The SST from REF_cmip, featured by a stronger cold tongue extending excessively westward along the equator, is warmer than the HadISST between 5°S and 20°S in the central and eastern Pacific. A band of cold SST bias down to −3 K between 5°S and 5°N across the Pacific can be clearly seen from the difference between them (Figure 8d). There is a conspicuous region of warm bias up to 2 K between 5°S and 20°S east of 150°W, which has been attributed to the underestimation of clouds in this region (e.g., Ma et al. 1996). It should be noted that

this study compares the SST from pre-industrial simulations to the present-climate observations. Considering the global warming trend, the pre-industrial SST should be colder than the 20th century observations, suggesting that the cold bias in cold tongue region may be overestimated and the warm bias in the central and eastern Pacific may be underestimated. When the new boundary-layer turbulence and shallow convection schemes are used, the simulated sea surface water is cooled down almost in the entire Pacific, especially in the SEP where the warm water in REF_cmip is cooled down by up to 4 K (Figure 8f).

It seems that the warm SST biases in REF_cmip are overcorrected in NEW_cmip by using new BL and shallow convection schemes, leading to a few degrees of cold bias in the SEP region. The area-averaged RMSE of SST in the tropical Pacific is 0.43 K in REF_cmip and actually deteriorates to 1.57 K in NEW_cmip. The common warm SST biases in CGCMs may come from several sources. Besides the underestimation of the shadowing effect due to a lack of stratocumulus that cover the SEP region, a poor representation of the oceanic surface cooling, by advection or mixing with the colder subsurface water, may also contribute to the warm biases (Richter, 2015). Also, some studies have found that shortwave radiation biases in marine stratocumulus regions are overcompensated for by excessive latent heat flux, which suggests a different origin of the warm SST biases (de Szoeke and Xie, 2008; Toniazzo and Woonough, 2014; Vanniere et al., 2014; Xu et al., 2014; Zheng et al., 2011). Recently, Hourdin et al. (2015) revealed that coupled models with warmer SST over the eastern tropical oceans present a lack of surface evaporative cooling in atmospheric simulations forced by SST. In the NEW_cmip simulation, an overestimation of the shadowing effect due to increased stratocumulus clouds may act to compensate for less surface evaporative cooling and make the sea surface cool enough to reduce precipitation in the SEP region.

The SST differences between the NEW_cmip and REF_cmip simulations are determined by both the net surface heat flux difference and ocean dynamic heat transport difference. The differences between NEW_cmip and REF_cmip runs for the atmospheric forcing on SST via surface heat flux $\Delta Q_{atm}$ is composed of differences of shortwave radiation $\Delta Q_{SW}$, longwave radiation $\Delta Q_{LW}$, latent heat flux $\Delta Q_{LH}$, and sensible heat flux $\Delta Q_{SH}$:

$$\Delta Q_{atm} = \Delta Q_{SW} + \Delta Q_{LW} + \Delta Q_{LH} + \Delta Q_{SH} \ . \tag{11}$$

The contributions of these four components are presented in Figure 9. In the SEP, the net atmospheric heat flux at sea surface is reduced in NEW_cmip, contributing to the cooled water in these regions. Among the atmospheric radiative heat flux components, the shortwave radiation is dominant, while the longwave radiation exerts a smaller and opposite effect. The latent heat flux is similar to the longwave radiative flux, and the sensible heat flux is negligible. Overall, the shortwave response explains most of net surface heat flux response in the SEP. In the SEP, where the Sc-to-Cu transition is prominent, the improved cloud simulation in NEW_cmip will enhance the shadowing effect due to increased stratocumulus, leading to oceanic surface cooling as seen in Figure 8f. According to the relationship between precipitation and SST, cooled SST cause reduced precipitation. This can explain why the excessive precipitation in the SEP is reduced and closer to the observation in NEW_cmip.

## 4.3 The Walker circulation and the surface wind stress

Changes in the simulated SSTs lead to changes in the simulated large-scale atmospheric circulation. The JRA-55 reanalysis data shows descending motion east of 150°W, which corresponds to the rainless state in the eastern Pacific between 5°S and 10°S, and intense ascending motion west of the dateline, which corresponds to the notable precipitation in the SPCZ (Figure 10a). In REF_cmip (Figure 10b), the weaker downward motion is limited to the middle and upper troposphere east of 125°W. The stronger upward motion occurs in the middle and upper troposphere between 150°W and 135°W, and is extended eastward to 100°W in the lower troposphere. Both the strengthened large-scale ascending motion and the weakened descending motion in the eastern part of the Walker circulation contribute to the excessive precipitation in the southern ITCZ region, resulting in the double ITCZ bias in REF_cmip. When the new boundary-layer turbulence and shallow convection schemes are used, the descending branch of the Walker circulation is better simulated in NEW_cmip (Figure 10c). Compared with REF_cmip, the upward motion in the low troposphere shrinks to 120°W in NEW_cmip. Furthermore, the downward motion in the middle and upper troposphere is enhanced and expanded westward to 140°W, making it in better agreement with the JRA-55 reanalysis. The improved descending branch of the Walker circulation is consistent with the decrease of precipitation relative to REF_cmip.

Figure 11 compares the annual mean surface wind stress vectors and surface convergence from REF_cmip and NEW_cmip simulations with JRA-55 reanalysis. In the eastern Pacific, the reanalysis shows convergence of northeasterly and southeasterly wind stresses in the northern ITCZ. The easterly and southeasterly wind stresses dominant central and eastern Pacific between 0° and 15°S, and no distinct convergence exists in these regions (Figure 11a). In the REF_cmip simulation (Figure 11b), the wind stress between 0° and 5°S is northeasterly compared to the observed easterlies, resulting in a convergence band in the central and eastern Pacific between 5°S and 10°S, which corresponding to the spurious southern ITCZ rainfall band. A prominent divergence zone also appears across the equatorial Pacific, which corresponds to the dry tongue in precipitation. The modified boundary-layer turbulence and shallow convection schemes result in increased southeasterly winds off the west coast of South America in NEW_cmip (Figure 11c). Specifically, the difference between NEW_cmip and REF_cmip clearly shows the strengthened southeasterly trade winds in the eastern Pacific between 5°S and 10°S (Figure 11d), corresponding to the stronger descending branch of the Walker circulation in NEW_cmip. Boundary layer convergence is primarily affected by SST gradients and can be usefully viewed as a forcing on deep convection over the tropical oceans (Back and Bretherton, 2009a, b). It is shown in Figure 10d that NEW_cmip produces relative divergence in the southern Pacific between 5°S and 15°S compared to REF_cmip, which corresponds to the eliminated southern ITCZ rainfall band resulting from weaker deep convection.

## 4.4 Enhanced cold advection in the upper ocean

Because of the strengthened southeasterly wind stress in and northwest of the SEP region, the south equatorial current in the upper ocean is enhanced. Figure 12 shows the longitude-depth cross section of zonal oceanic current and temperature averaged over 5°S – 10°S for the difference between NEW_cmip and REF_cmip. Compared with REF_cmip, the climatological

westward zonal current in NEW_cmip over 5°S−10°S is enhanced by more than 8 cm/s above 120 m over the central to eastern Pacific. Further analysis indicates that the simulated subsurface temperature is reduced by more than 2 K above 80 m east of 135°W in NEW_cmip. Apparently, the enhanced westward ocean current over the whole zonal band helps transport cooler water from east to west and prevents the warm water in the western Pacific from extending eastward in NEW_cmip.

## 5 Discussion

### 5.1 The Sc-to-Cu transition

The relative contribution of the UWMT boundary-layer scheme and modified Hack shallow convection scheme in portraying the Sc-to-Cu transition is further examined. Figure 13 presents the vertical cross sections of cloud fraction through the five main subtropical stratocumulus regions in the sensitivity experiments of UWMT_amip and mHack_amip. UWMT_amip reproduces the elevated cloudiness maxima, which can be expected from the applicability of the UWMT turbulence scheme in simulating stratocumulus, but still suffers from the rapid dissipating of cloud cover downstream off the coasts (Figure 13, left). On the contrary, mHack_amip produces the gradual reduction of cloud amount offshore but misses the elevated cloudiness maxima, and constrains the cloud to a very low level (Figure 13, right). Neither UWMT_amip nor mHack_amip captures all the significant transition characteristics, indicating that the coupling between boundary-layer turbulence and shallow convection is important for an accurate simulation of the Sc-to-Cu transition.

In this study, introducing the decoupling criterion in the Hack scheme aims to better treat the transition between the stratocumulus regime and shallow cumulus regime. Regime-dependent parameterizations have limitations when representing various cloud regimes and their transitions. Two classes of unified parameterizations of boundary-layer turbulence and shallow convection have been documented in the literature, known as the eddy diffusion mass flux (EDMF) approach (Siebesma et al., 2007; Pergaud et al., 2009; Hourdin et al., 2013), and the higher-order turbulence closure (HOC) approach (Bogenschutz et al., 2013; Guo et al., 2010, 2014). Effects of these greater unification of cloud parameterizations on the double ITCZ bias in CGCMs will be further explored in future work.

### 5.2 Robustness of the alleviated double ITCZ

More investigations on the role of boundary-layer turbulence and shallow convection schemes in the double ITCZ formation in different CGCMs are desired. For simplicity, the UWMT and modified Hack schemes are employed in the high-resolution Beijing Climate Center Climate System Model version 2 (BCC-CSM2-HR), which is largely different from BCC-CSM2-MR with respect to model physics and dynamics, to examine the robustness of the alleviated double ITCZ through improving parameterizations of boundary-layer turbulence and shallow convection.

During the transition from BCC-CSM2-MR to BCC-CSM2-HR, the atmospheric component increased its horizontal resolution from T106 (~ 1.125°) to T266 (~ 0.45°) with a higher model top, and the physics package was essentially updated,

especially the deep convection scheme. Furthermore, the oceanic component was upgraded to the Modular Ocean Model version 5 (MOM5). However, previous versions of BCC-CSM2-HR suffered from the double ITCZ syndrome until the UWMT and modified Hack schemes were introduced. Before improving parameterizations of boundary-layer turbulence and shallow convection, BCC-CSM2-HR simulated a southern rainfall band with excessive eastward extension over the central and eastern Pacific and two nearly parallel rain belts over the equatorial Atlantic (Figure 14a). This suggests that the boundary-layer and shallow convection schemes contribute primarily to the double ITCZ bias in BCC-CSM2-HR. The tropical precipitation patterns simulated in the frozen version of BCC-CSM2-HR, which is equipped with new boundary-layer turbulence and shallow convection schemes, barely manifest a double ITCZ, as shown in Figure 14b. The triangular-shaded dry region in the SEP reproduced by BCC-CSM2-HR resembles the observed much better than that simulated in the revised BCC-CSM2-MR, probably due to the improved interactions among the boundary-layer turbulence, shallow convection, and other processes. Anyway, improving parameterizations of boundary-layer turbulence and shallow convection shows robustness in mitigating the double ITCZ syndrome in different BCC coupled models.

## 6 Summary and conclusions

It is a challenge to eliminate the double ITCZ problem, one of the most prominent systematic biases in CGCMs. This study investigates the roles of BL turbulence and shallow convection parameterizations in alleviating the double ITCZ bias using the BCC-CSM2-MR. The original BCC-CSM2-MR presents a serious double ITCZ problem in precipitation, with two significant zonal rain bands straddling the equator across the Pacific. In contrast, the revised BCC-CSM2-MR with new configurations of BL turbulence and shallow convection schemes remarkably alleviates the spurious southern ITCZ bias associated with SST warm bias in the SEP, owing to the reduced net surface shortwave radiation associated with the increased low cloud fraction. Correspondingly, the cooler water in the SEP induces stronger and wider subsiding motion of the Walker circulation. The stronger Walker circulation and enhanced eastward surface wind stress in turn lead to increased oceanic zonal cold advection from east to west in the southern equatorial Pacific.

The double ITCZ is a prominent bias in CGCM, which may result from discrepancies in the representation of both atmospheric and oceanic processes (Zhang et al., 2019; Song and Zhang, 2020). Consistent with previous studies (Ma et al., 1996; Yu and Mechoso, 1999; Dai et al., 2003, 2005; Qin and Lin, 2018), this study emphasizes the importance of stratus clouds and SSTs in the SEP in alleviating the double ITCZ bias via changes in the tropical circulation and ocean dynamics. However, different with studies with modifications of cloud scheme itself, this study focuses on the roles of BL turbulence and shallow convection in the low-level cloud simulation, and the subsequent impacts on the mitigation of the double ITCZ. It is found that the simulated low clouds are sensitive to both BL turbulence and shallow convection parameterizations. Better consistency between the BL turbulence scheme and the shallow convection scheme results in increased low-level clouds and their shortwave radiative forcing. The exaggerated cloud amount and cloud effect can be explained as a compensation for

related biases in other physical processes. In conclusion, this study indicates that improving parameterizations of BL turbulence and shallow convection is an effective way to reduce the double ITCZ in CGCMs.

The BL processes can not only affect SST by changing the stratocumulus and its radiative effect, but also control the surface evaporative cooling by convective transport of humidity at the surface and then SST (Hourdin et al., 2020). Using a mass flux representation of the organized structures of the convective BL coupled to eddy diffusion, Hourdin et al. (2020) showed that an increased near-surface drying led to a reduction of the warm bias in the eastern tropical oceans in the Institute Pierre Simon Laplace coupled model, IPSL-CM6A. They concluded that a good representation of BL convection is required

to maintain a strong contrast between trade winds cumulus regions and stratocumulus regions. Similarly, this study adopts the eddy diffusion mass flux (EDMF) approach, which seeks to unify BL and shallow convective processes by the marriage of UWMT and modified Hack schemes. However, there are still large discrepancies in the simulated Sc-to-Cu transition compared to observations, as shown in Figure 3, which suggests that parameterization of BL convection should be further improved. Moreover, the role of surface evaporative cooling needs to be explored when improving representation of BL

convection.

      In the present study, the alleviation of the double ITCZ problem is accompanied by an amplification of the cold tongue bias, as found by many early efforts focusing on the role of low clouds. The inverse responses of the double ITCZ and cold tongue biases to southeast Pacific low clouds suggest that other parameterized processes, e.g., deep convection, may play an important role in the accurate simulation of the ITCZ-cold tongue complex. Song and Zhang (2018) demonstrated that the

515 double ITCZ bias is largely eliminated and the cold bias in equatorial cold tongue is also significantly reduced through modifying the deep convection scheme in the Community Earth System Model version 1.2.1 (CESM1.2.1). In the high-resolution BCC-CSM2-HR, the cold tongue simulation seems to be unaffected by alleviating the double ITCZ bias, which may benefit from the improved deep convection parameterization. The effect of improving the representation of deep convection on the double ITCZ bias in BCC-CSM2-MR is beyond the scope of this paper and will be explored in future work.

*Code and data availability.* The source codes of BCC-CSM2-MR, model input data files, and scripts to reproduce the simulations that are used in this study have been archived and made publicly available for downloading from http://doi.org/10.5281/zenodo.3940326, as are model output data to produce the plots for all the simulation experiments presented in the paper.

*Author contributions.* YL modified the source codes, designed and performed all the experiments presented in the paper, and wrote a large part of the paper. TW supervised the BCC-CSM development and provided critical comments on the paper. All the authors continuously discussed the model development and the results.

*Competing interests.* The authors declare that they have no conflict of interest.

*Acknowledgements.* This work was supported by The National Key Research and Development Program of China (2016YFA 0602100). We gratefully acknowledge the groups of GPCP (https://www.esrl.noaa.gov/psd/data/gridded/data.gpcp.html), CM AP (https://www.esrl.noaa.gov/psd/data/gridded/data.cmap.html), CALIPSO-GOCCP (http://climserv.ipsl.polytechnique.fr/c fmip-obs), CERES-EBAF (https://ceres.larc.nasa.gov/products.php?product=EBAF-Product), HadISST (https://www.metoffi ce.gov.uk/hadobs/hadisst), and the JRA-55 reanalysis (https://jra.kishou.go.jp/JRA-55) for providing public access to various observational data sets. All the graphics in this study are created by the NCAR Command Language (NCL; doi:10.5065/D6 WD3XH5). The authors thank the three anonymous reviewers and the editor for their valuable comments that helped to impr ove the manuscript.

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

**Table 1.** Summary of experimental setups.

| Run name | Boundary layer scheme | Shallow convection scheme |
|---|---|---|
| REF_amip | HB | Hack |
| NEW_amip | UWMT | modified Hack |
| REF_cmip | HB | Hack |
| NEW_cmip | UWMT | modified Hack |
| UWMT_amip | UWMT | Hack |
| mHack_amip | HB | modified Hack |

**Table 2.** Evaluation of the low-level cloud fraction (%) from REF_amip and NEW_amip simulations against GOCCP observations. Shown are the area-averaged biases and root-mean-square errors (RMSEs) between simulated and observed low-level cloud amounts over the globe, in the tropics and for the five main subtropical marine stratocumulus regions, which is indicated in Figure 2. Pattern correlations are calculated for the global and tropical low-level cloud distribution in the simulations, respectively.

| Region | Bias | | RMSE | | Pattern Correlation | |
|---|---|---|---|---|---|---|
| | REF_amip | NEW_amip | REF_amip | NEW_amip | REF_amip | NEW_amip |
| Global | -12.48 | -8.35 | 12.57 | 8.49 | 0.76 | 0.84 |
| Tropical | -14.18 | -8.64 | 14.26 | 8.74 | 0.72 | 0.89 |
| Peruvian | -35.73 | -7.63 | 36.91 | 14.89 | | |
| Californian | -32.61 | -22.40 | 33.69 | 24.80 | | |
| Australian | -38.43 | -11.41 | 39.56 | 18.81 | | |
| Namibian | -28.37 | -3.45 | 30.11 | 12.55 | | |
| Canarian | -12.56 | 6.03 | 15.95 | 20.68 | | |

**Table 3.** Evaluation of the precipitation rate (mm day$^{-1}$) from REF_cmip and NEW_cmip simulations against GPCP and CMAP observational estimates. Shown are the area-averaged biases and root-mean-square errors (RMSEs), and pattern correlations between simulated and observed precipitation rate in the tropical Pacific (30°S−30°N, 120°E−90°W).

| Observational Data | Bias | | RMSE | | Pattern Correlation | |
|---|---|---|---|---|---|---|
| | REF_cmip | NEW_cmip | REF_cmip | NEW_cmip | REF_cmip | NEW_cmip |
| GPCP | 0.89 | 0.44 | 0.94 | 0.54 | 0.78 | 0.81 |
| CMAP | 0.33 | -0.12 | 0.48 | 0.36 | 0.80 | 0.81 |


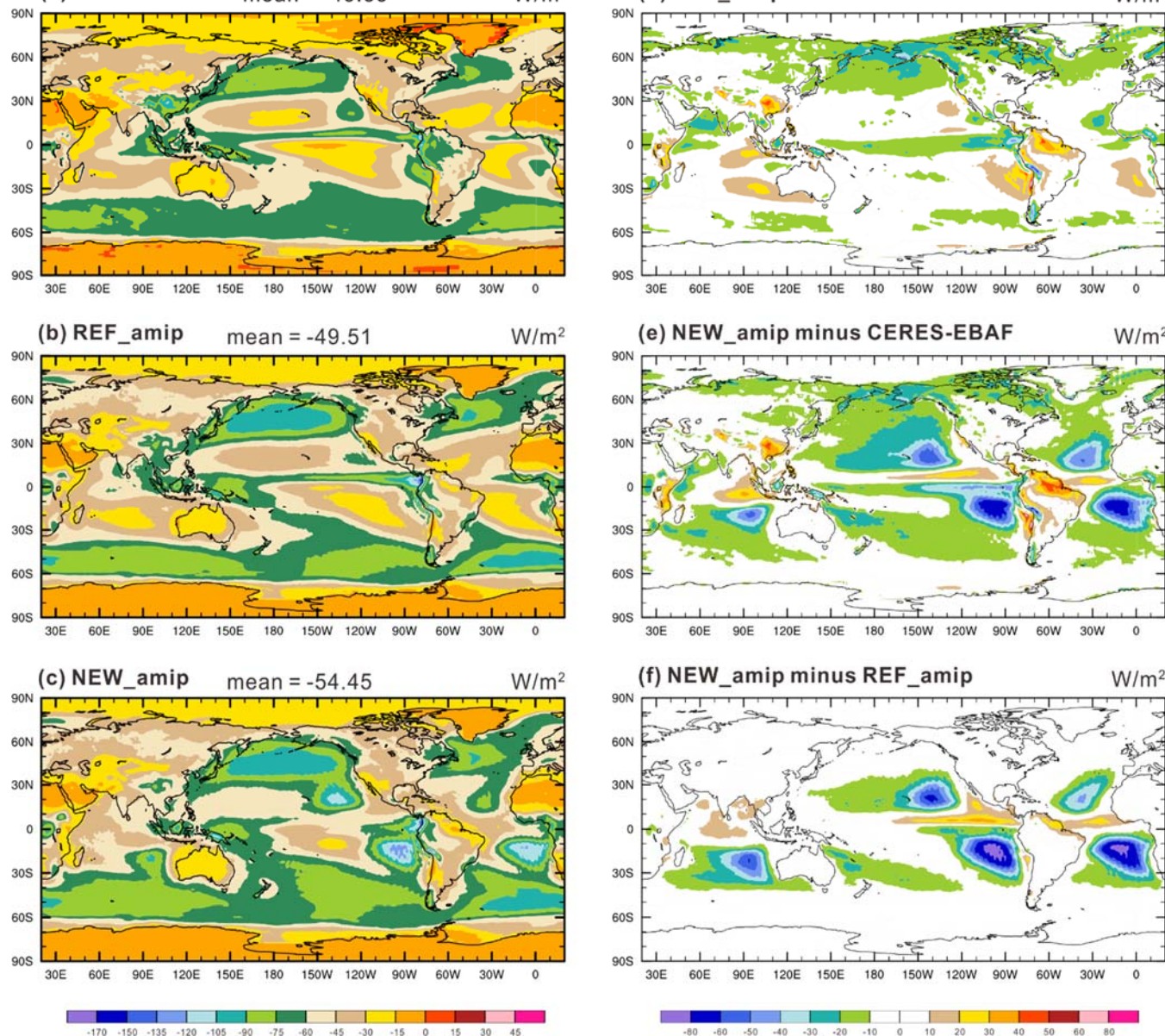

**Figure 1.** Annual mean climatologies of TOA SWCRF (W m$^{-2}$) from (a) CERES-EBAF observations (from 2001 to 2010), (b) REF_amip, (c) NEW_amip, and the differences between (d) REF_amip and observations, (e) NEW_amip and observations, (f) NEW_amip and REF_amip. Shown atop panels (a), (b) and (c) are global mean values.

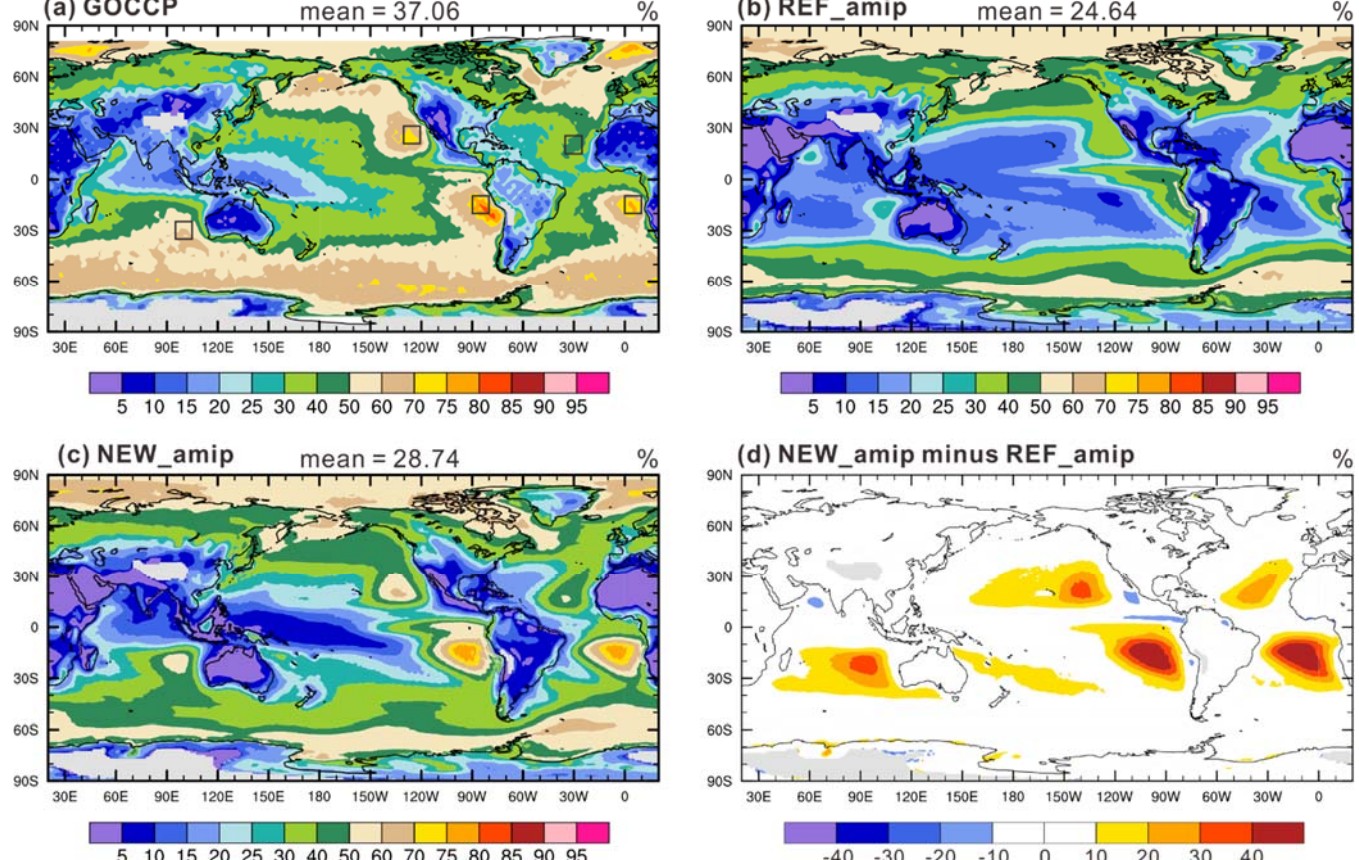

**Figure 2.** Annual mean low-level cloud fraction (%) from (a) GOCCP (from 2007 to 2012), (b) REF_amip, (c) NEW_amip, and (d) the difference between NEW_amip and REF_amip. Shown atop panels (a), (b) and (c) are global mean values. Boxes in panel (a) indicate the five main subtropical marine stratocumulus regions, namely, Peru (10°-20°S, 80°-90°W), California (20°-30°N, 120°-130°W), Australia (25°-35°S, 95°-105°E), Namibia (10°-20°S, 0°-10°E), and Canaria (15°-25°N, 25°-35°W).


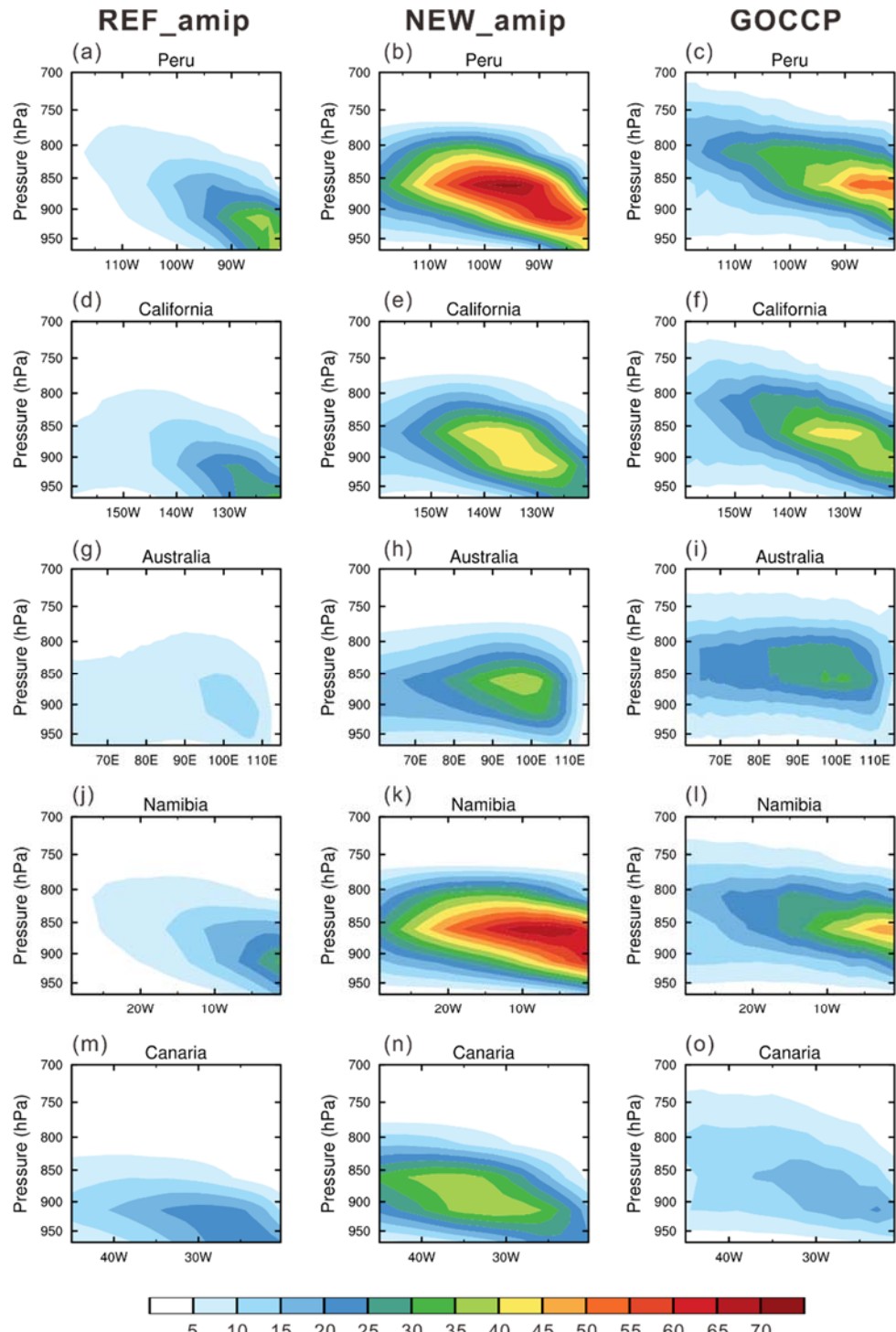

**Figure 3.** Cross sections of cloud fraction from five locations for (left) REF_amip, (center) NEW_amip, and (right) GOCCP. Refer to the boxes in Fig. 2 for the locations of the cross sections.

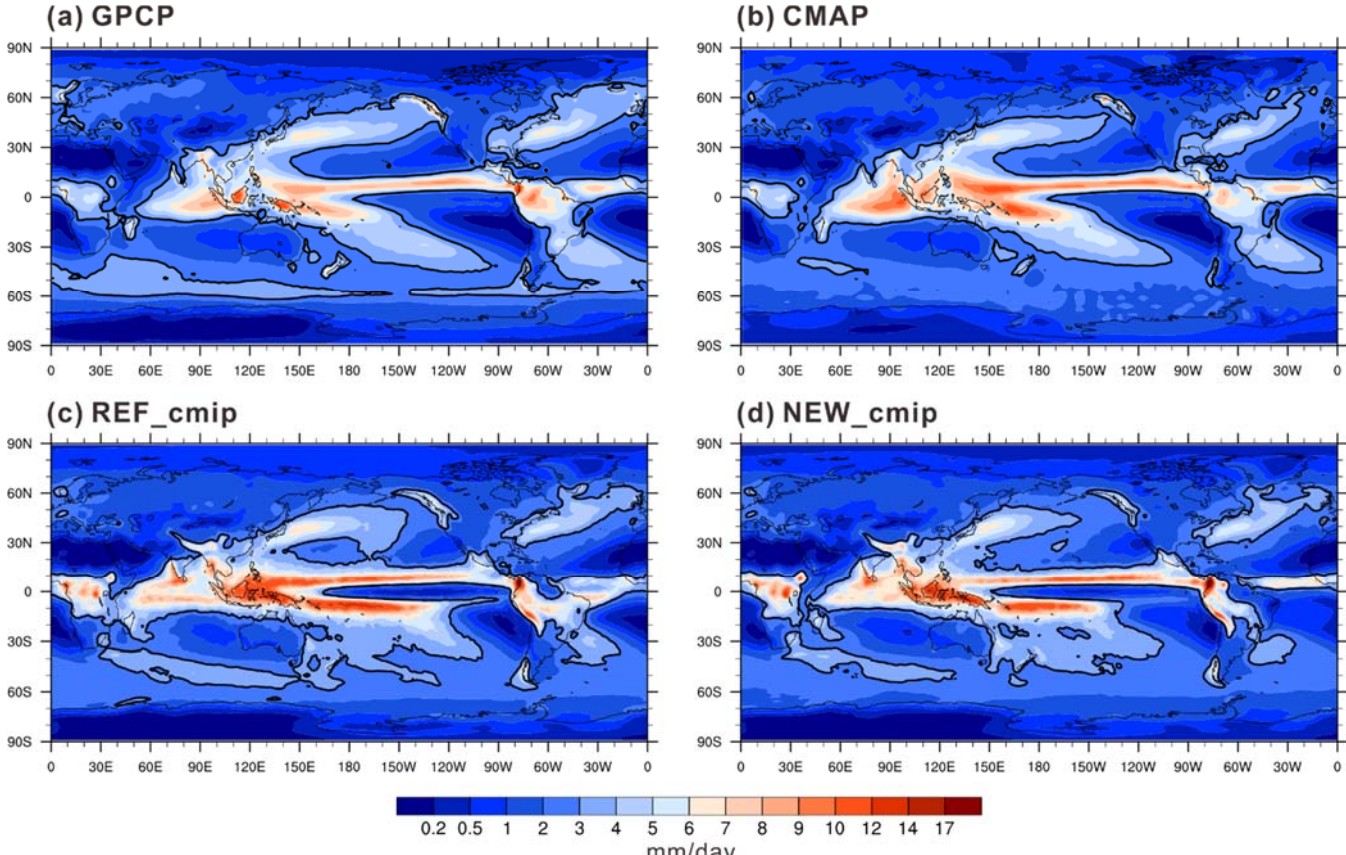

**Figure 4.** Annual mean precipitation rate (mm day$^{-1}$) from (a) GPCP, (b) CMAP, (c) REF_cmip, and (d) NEW_cmip. The 3 mm day$^{-1}$ contour is included in bold for reference.


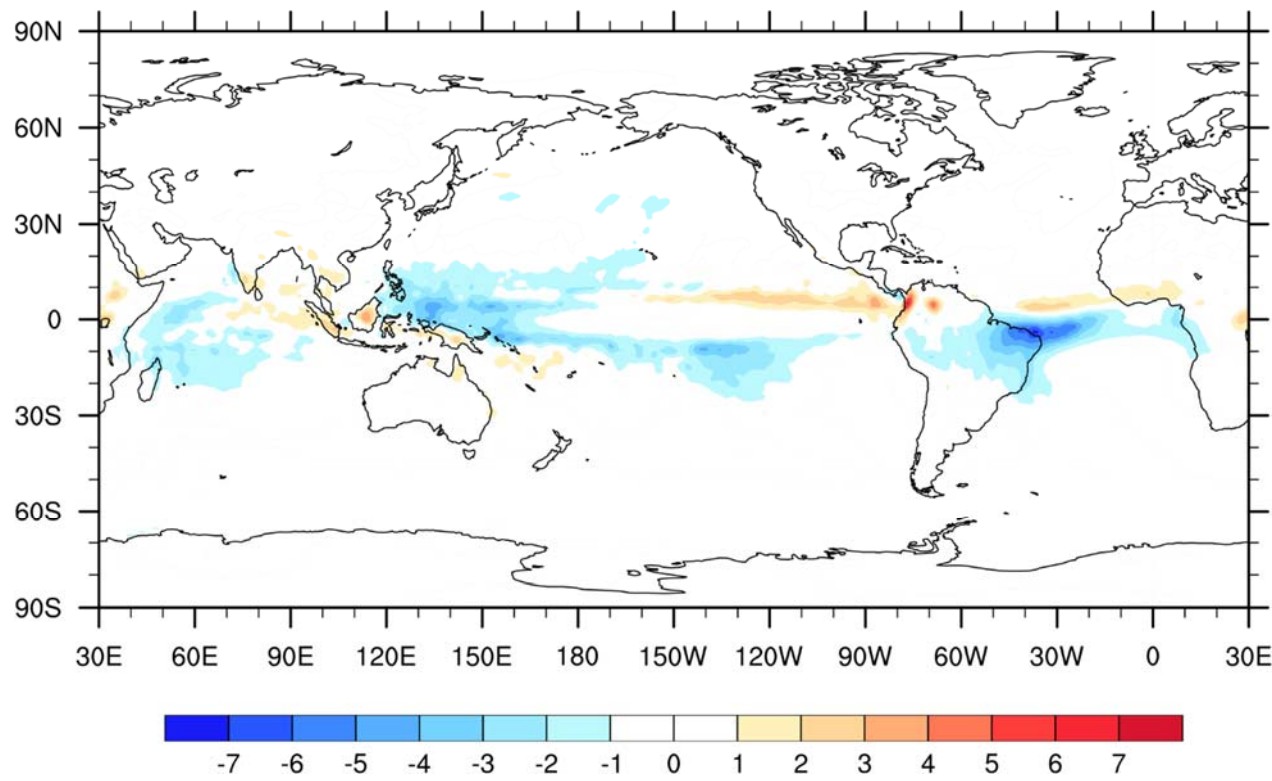

**Figure 5**. Differences of annual-mean precipitation rate (mm day$^{-1}$) between NEW_cmip and REF_cmip.

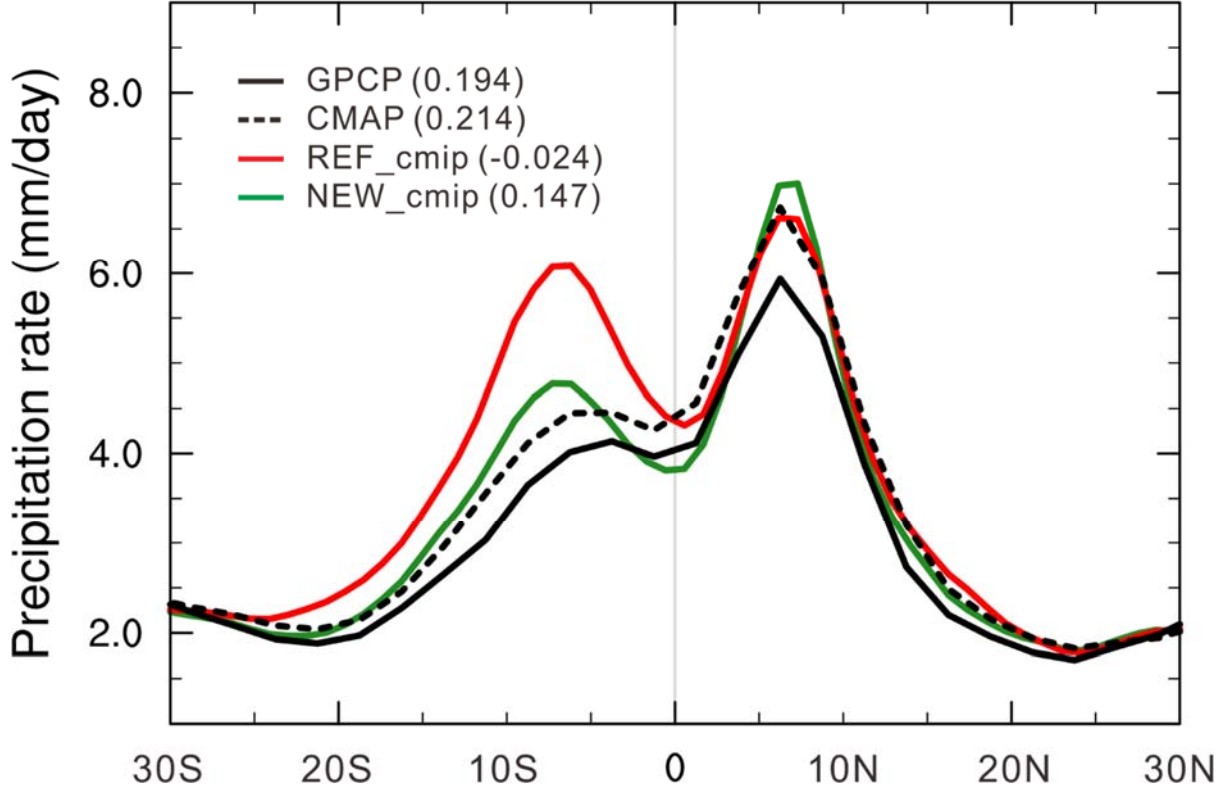


**Figure 6.** Zonal mean precipitation rate (mm day⁻¹) from GPCP, CMAP, REF_cmip and NEW_cmip in the tropics. Values of the tropical precipitation asymmetry index are indicated in parentheses.

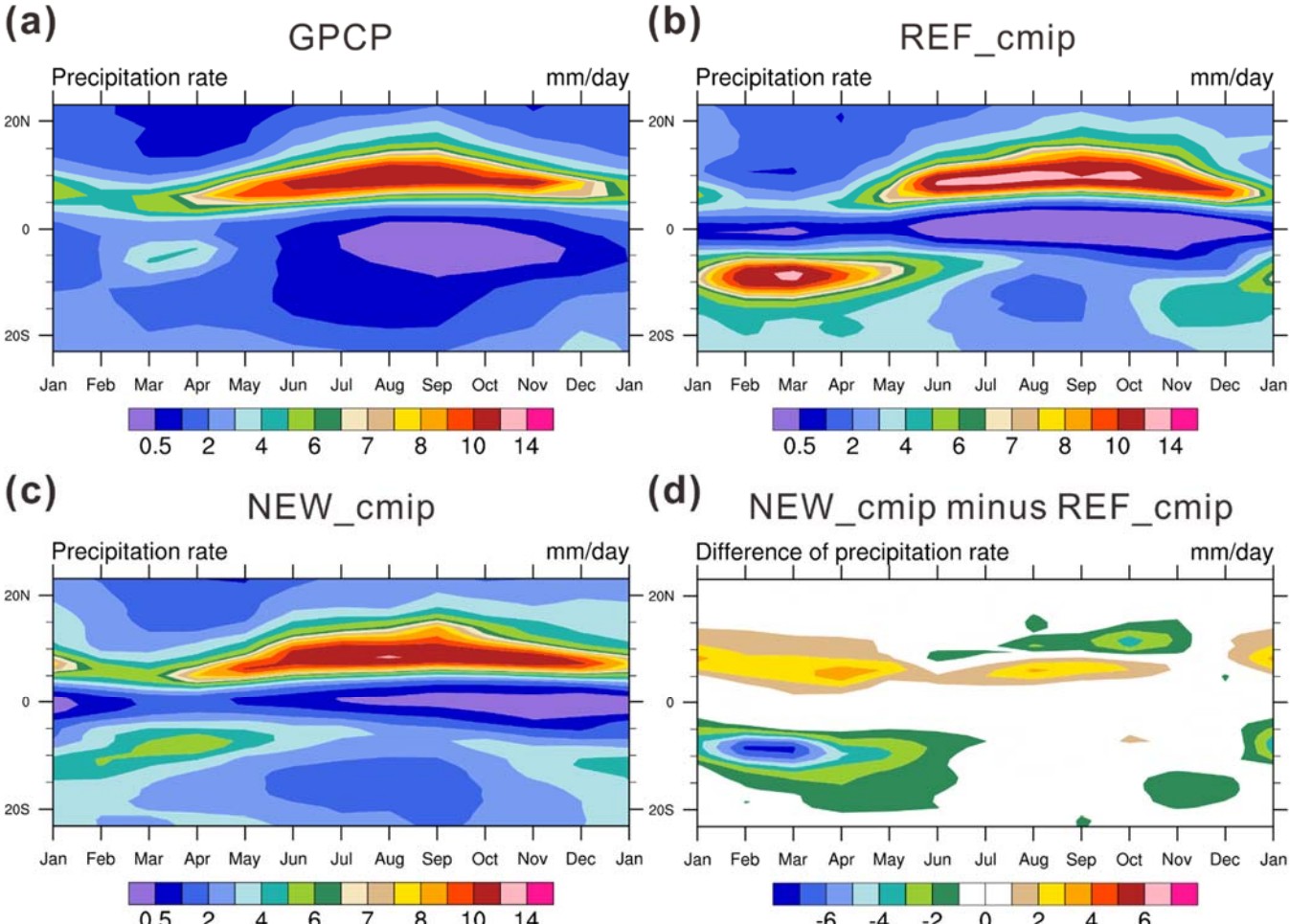

**Figure 7.** Seasonal cycle of precipitation rate (mm day$^{-1}$) averaged over eastern Pacific (90° - 160°W) for (a) GPCP, (b) REF_cmip, (c) NEW_cmip, and (d) the difference between NEW_cmip and REF_cmip.

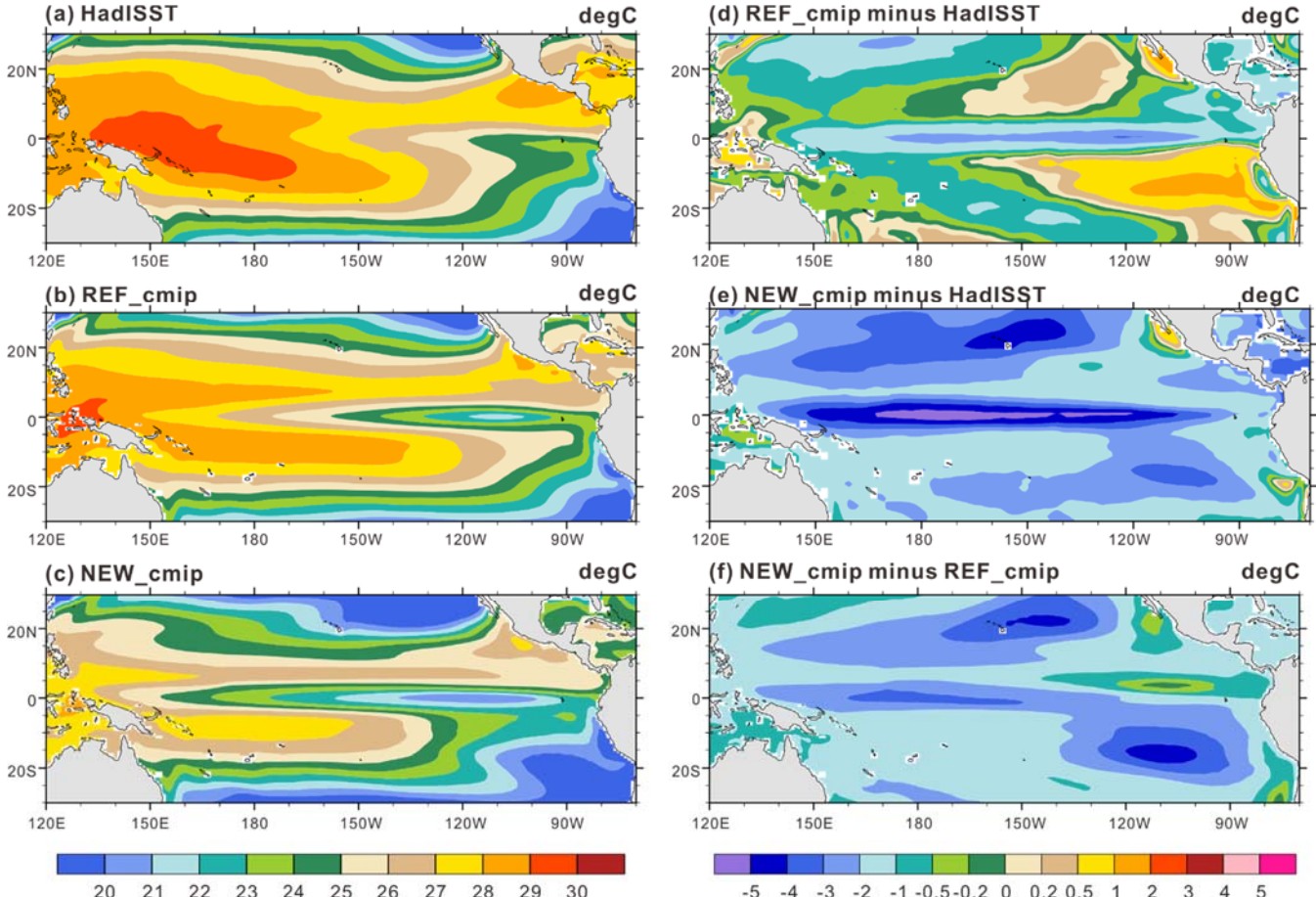

**Figure 8.** Annual mean sea surface temperature (°C) from (a) HadISST, (b) REF_cmip, and (c) NEW_cmip, and the difference between (d) REF_cmip and HadISST, (e) NEW_cmip and HadISST, and (f) NEW_cmip and REF_cmip.

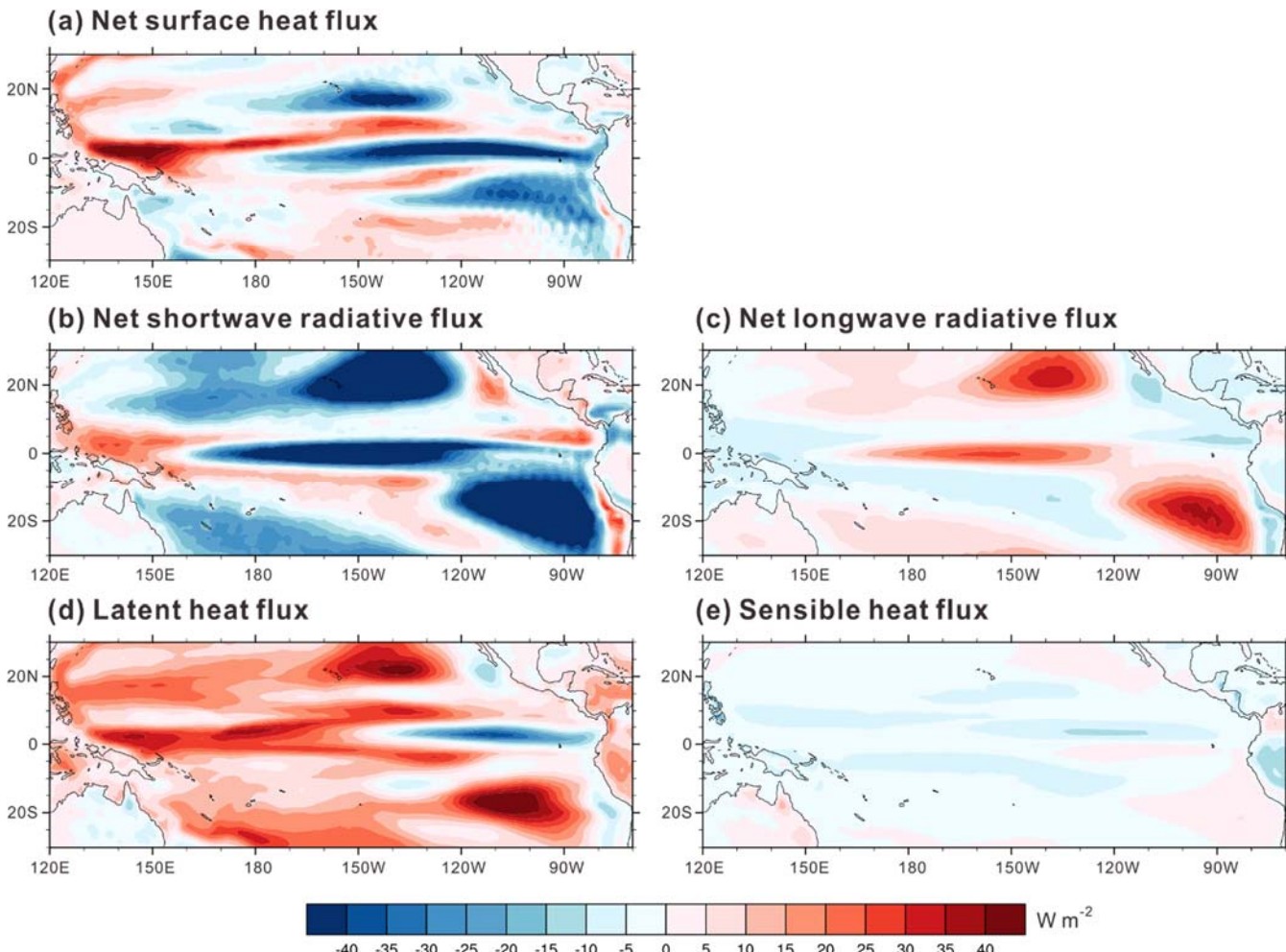

**Figure 9.** The differences between the NEW_cmip and REF_cmip simulations (W m$^{-2}$) for (a) net surface heat flux $\Delta Q_{atm}$, (b) net shortwave radiative flux $\Delta Q_{SW}$, (c) net longwave radiative flux $\Delta Q_{LW}$, (d) latent heat flux $\Delta Q_{LH}$, and (e) sensible heat flux $\Delta Q_{SH}$.


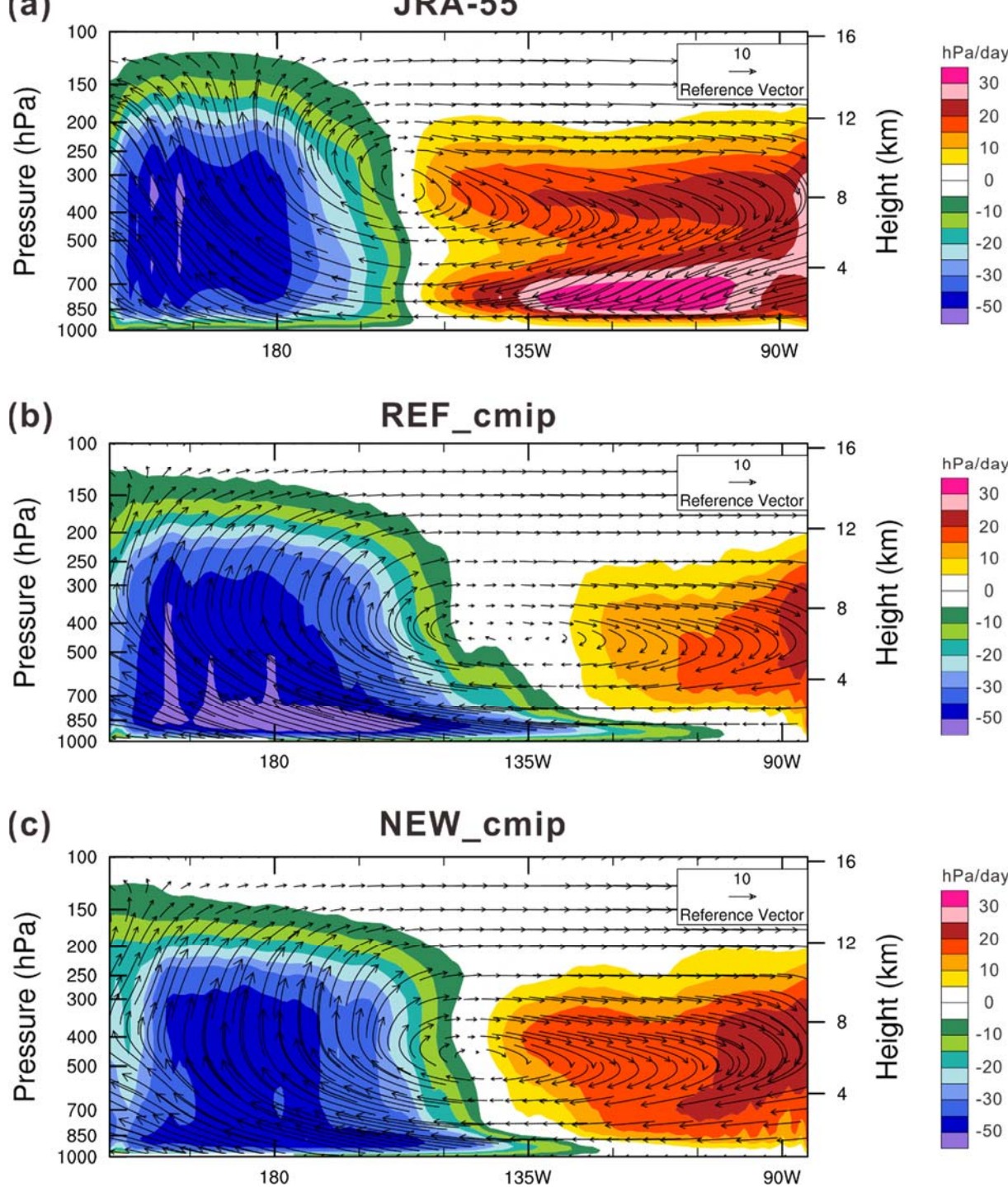

**Figure 10.** Annual mean vertical pressure velocity (hPa day$^{-1}$; shaded) and wind vectors (arrows) in the longitude-height cross section averaged over 5°S – 10°S for (a) the JRA-55 reanalysis, (b) REF_cmip, and (c) NEW_cmip.

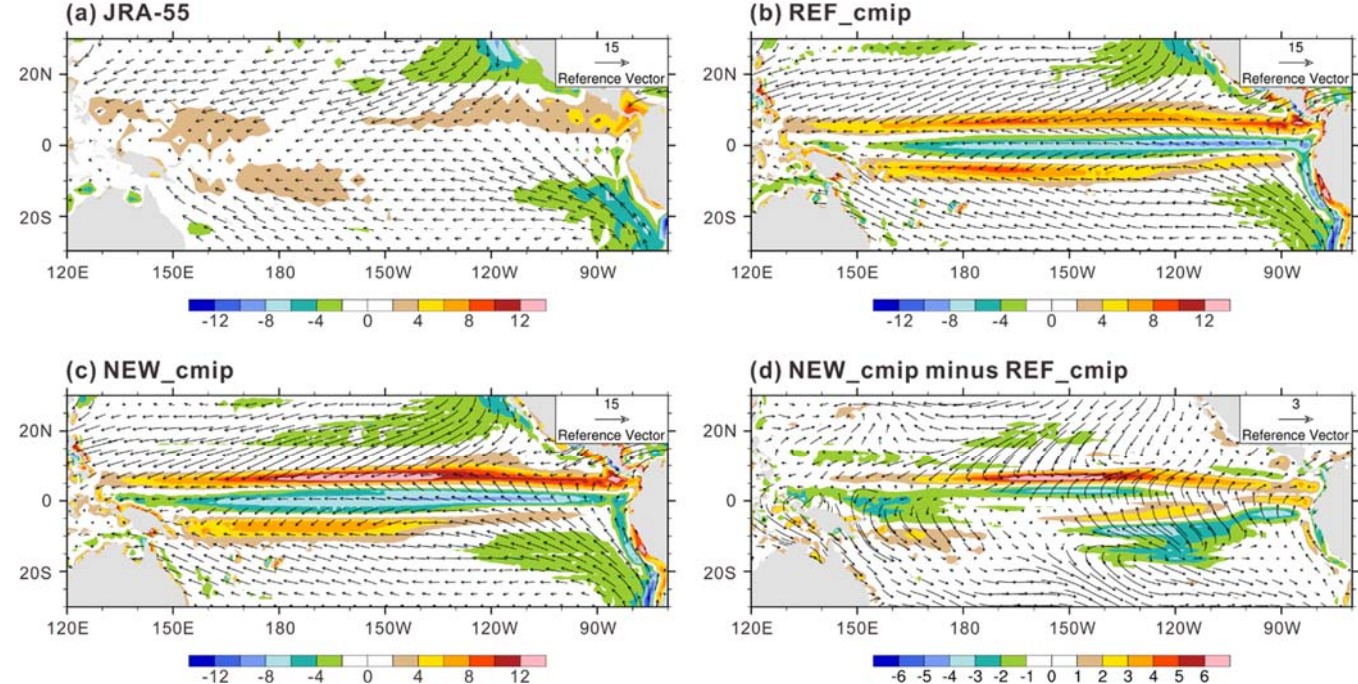


**Figure 11.** Annual mean wind stress vector and surface convergence (shaded, ×10⁻⁶ s⁻¹) from (a) JRA-55 reanalysis, (b) REF_cmip, (c) NEW_cmip, and (d) the difference between NEW_cmip and REF_cmip.

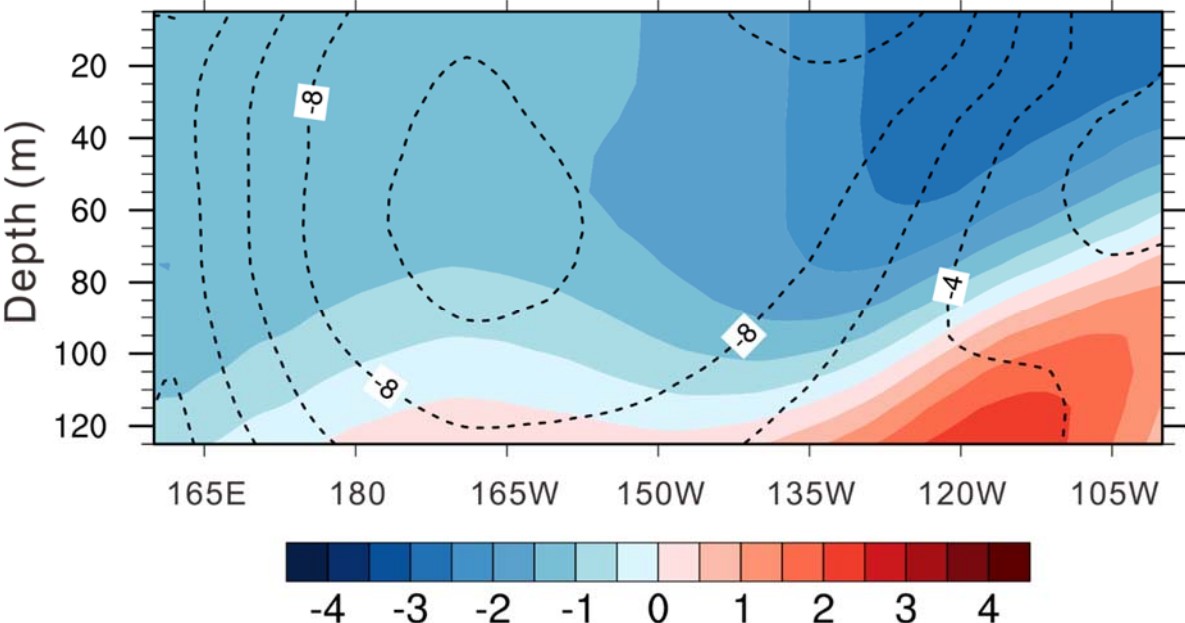

**Figure 12.** Longitude-depth cross section of zonal ocean current (contours; cm s[-1]) and temperature (shaded; °C) averaged over over 5°S – 10°S for the difference between NEW_cmip and REF_cmip.

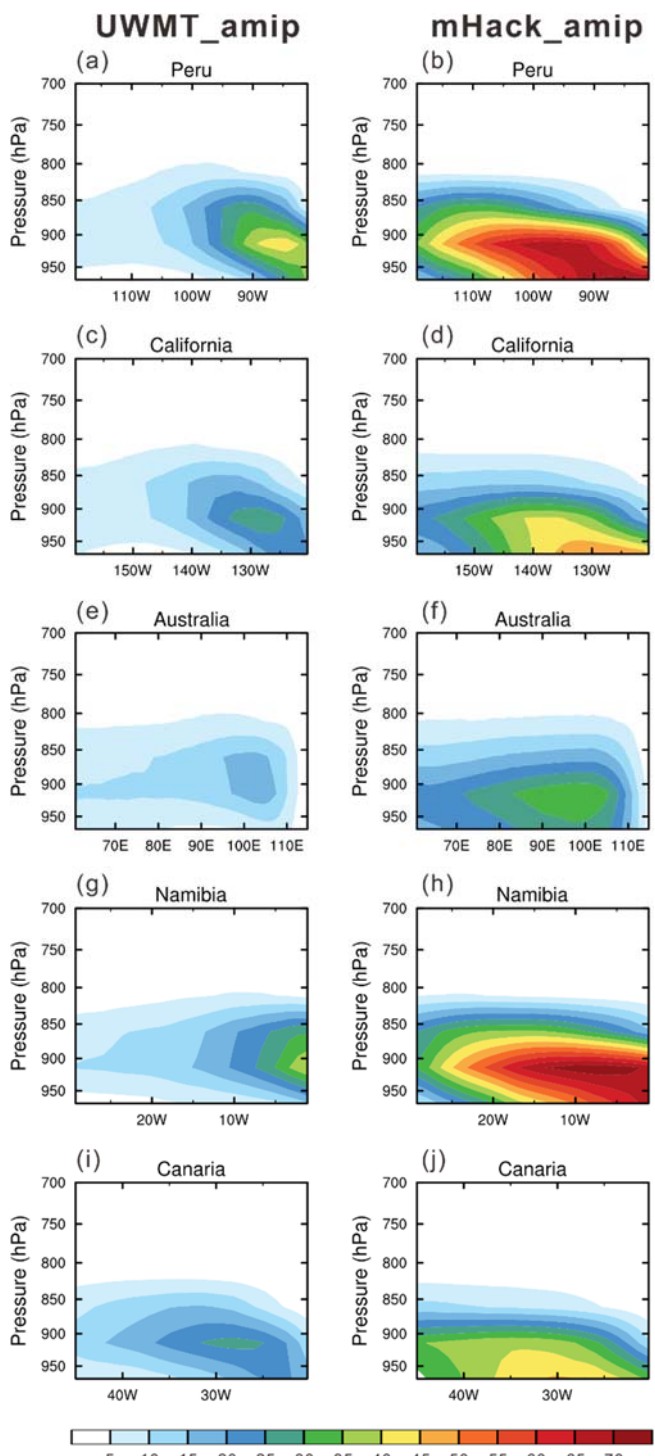

**Figure 13.** Cross sections of cloud fraction from five locations for (left) UWMT_amip, and (right) mHack_amip. Refer to the boxes in Fig. 2 for the locations of the cross sections.

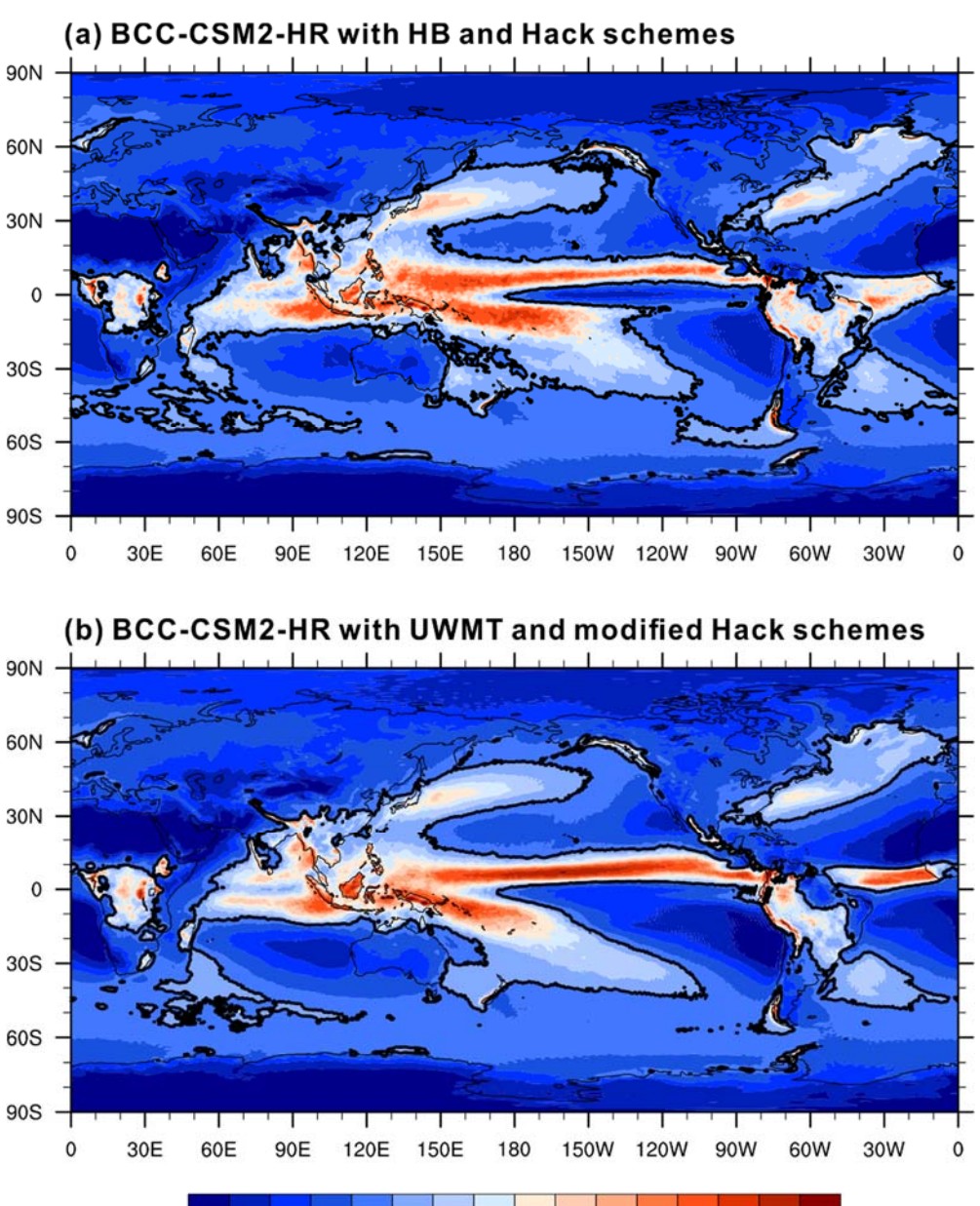

**Figure 14.** Annual mean precipitation rate (mm day$^{-1}$) from (a) intermediate version of BCC-CSM2-HR with original boundary-layer turbulence and shallow convection schemes, and (b) frozen version of BCC-CSM2-HR with new boundary-layer turbulence and shallow convection schemes. The 3 mm day$^{-1}$ contour is included in bold for reference.
