# Peer review of "Mitigation of the double ITCZ syndrome in BCC-CSM2-MR through improving parameterizations of boundary-layer turbulence and shallow convection"

_Geoscientific Model Development, 2020_

## Referee Comment (RC1) · Anonymous Referee #1 · 19 Mar 2020

Review of

**Mitigation of the double ITCZ syndrome in BCC-CSM2-MR through improving parameterizations of boundary-layer turbulence and shallow convection**

Yixiong Lu, Tongwen Wu, Yubin Li, & Ben Yang

**General**

The authors report the improvement in the known 'double ITCZ bias', common in most coupled climate models, in the BCC-CSM2-MR model. The authors aptly report the improvement caused by the implementation of new boundary layer and shallow cloud parameterisations. I am not an expert in the associated parameterizations, and so not in a position to critically assess the technical and physical aspects related to their implementation. I do have some comments with regards to the reported improvement in the double ITCZ bias and on the general presentation of the analysis. I find the work to be generally well presented and within the scope of GMD. With regards to the mitigation of the double ITCZ bias, I encourage the authors to take into consideration the comments provided below.

**Specific comments**

1. Adding the values of critical parameters used in the revised parameterization would help the readers and enable the reproducibility of the reported results (e.g., $A$ in Eq. 5).
2. Line 27: the double ITCZ bias is seen year-round in the central and western Pacific, but only during the SH rainy season over the eastern Pacific (and Atlantic). (See Adam et al. 2018 and Li & Xie 2014)
3. Lines 228-230: increased cloud fraction in the subtropical eastern Pacific cools surface waters which subduct and eventually end up in the cold tongue. It is worth mentioning this coupling mechanism here, which was shown to have an important effect by Burls et al. (2017).
4. Lines 255-256: both the surface temperature and surface temperature gradient have an important effect on boundary layer and deep convection, as shown by Back and Bretherton (2009). The effect of BL convergence by SST gradients is not accounted for in the analysis.
5. Lines 272-274: Indeed, the biases in the eastern Pacific and Atlantic are reduced. However, a negative bias in the Equatorial Indian Ocean seems to get worse. Are

the biases in the Indian Ocean, as well as the changes in the revised model, also related to the BL parameterization?

6.  Fig. 4: The cold tongue bias seems to persist in the revised model. (This is mentioned lated in the analysis.) Since the cold tongue bias is known to be closely linked to the double ITCZ bias, it is interesting and worth highlighting that an improvement is achieved only in one aspect of the bias.

7.  Lines 286-293: Both the anti-symmetric and symmetric components of the precipitation bias are significant (e.g., Adam et al. 2016). I suspect that the equatorial precipitation index (Adam et al. 2016, 2018), which was found to be strongly correlated with other phenomena (Popp and Lutsko 2017), will be quite different from observations in the revised model, in particular since equatorial precipitation in Fig. 5 is lower than observed.

8.  Section 4.4 and Fig. 11: This is an odd Figure. According to classic theory, wouldn't it be the curl of the wind stress that affects the zonal ocean currents, rather than the intensity of the Walker circulation? In any case, Fig. 11 and the short treatment of this potentially important aspect seem perfunctory. I would suggest either omitting this section from the paper or providing a more detailed and complete analysis.

9.  5.2 Indeed the HR model seems to dramatically improve the representation of tropical precipitation. However the reader is left curious and confused. The authors claim that it is the UWMT that accounts fo the major improvement in the HR model but provide virtually no support for this claim.

10. Lines 410-411: is this true also for the HR model?

**Technical/editorial comments**

14        promotes —> ameliorates

32        fake —> spurious

37        impediment to what?

49        convection and cloud radiative effects

62        Previous attempts

76        accounts for —> alleviates

**References**

-  Burls, N.J., Muir, L., Vincent, E.M. *et al.,* 2017*:* Extra-tropical origin of equatorial Pacific cold bias in climate models with links to cloud albedo. *Clim Dyn* **49,** 2093–2113

- Back, L.E. and C.S. Bretherton, 2009: A Simple Model of Climatological Rainfall and Vertical Motion Patterns over the Tropical Oceans. *J. Climate*, **22**, 6477–6497
- Adam, O., T. Schneider, F. Brient, and T. Bischoff, 2016: Relation of the double-ITCZ bias to the atmospheric energy budget in climate models, *Geophys. Res. Lett.*, ***43,*** 7670–7677
- Adam, O., Schneider, T., and Brient, F., 2018: Regional and seasonal variations of the double-ITCZ bias in CMIP5 models, *Clim. Dynam.*, **51**, 101-117
- Popp, M., & Lutsko, N. J., 2017: Quantifying the zonal-mean structure of tropical precipitation. *Geophysical Research Letters*, ***44***, 9470–9478
- Li, G. and Xie S.-P., 2014: Tropical biases in CMIP5 multimodel ensemble: The excessive equatorial Pacific cold tongue and double ITCZ problems. *J. Climate*, **27**, 1765-1780

---

## Short Comment (SC1) · 10 Jul 2020

This is an executive editor comment highlighting the ways in which this manuscript is not currently compliant with GMD policy on code and data availability.

This manuscript currently indicates that code and data is available on request from the author. This does not comply with GMD policy on code and data availability, and the manuscript must be rejected unless this can be remedied. In particular, code and data must be archived and published at the time of submission in order to facilitate the open

review process.

Please immediately archive and publish the code and data on which this manuscript depends and post the citations in a reply to this comment. Naturally, the code and data must be properly cited in the revised manuscript.

The full details of the code and data availability requirements are in the GMD model code and data policy: https://www.geoscientific-model-development.net/about/code_ and_data_policy.html. The reasons for the policy and more detail are provided in this editorial: https://doi.org/10.5194/gmd-12-2215-2019.

―――――――――――――――――――――

---

## Author Comment (AC1) · 11 Jul 2020

Dear editor,

Thank you for your comment on the ways in which this manuscript is not currently compliant with GMD policy on code and data availability.

This problem has been remedied. We have archived and published the code and data on which this manuscript depends on Zenodo (http://doi.org/10.5281/zenodo.3940326). Correspondingly, we have rewritten the

section of 'Code and data availability' in the revised manuscript, as follows,

Code and data availability. The source codes of BCC-CSM2-MR, model input data files, and scripts to reproduce the simulations that are used in this study have been archived and made publicly available for downloading from http://doi.org/10.5281/zenodo.3940326, as are model output data to produce the plots for all the simulation experiments presented in this paper.

Best regards,

Yixiong Lu and all co-authors

---

## Author Comment (AC2) · 7 Aug 2020

Dear Referee #1,

We would like to thank you for your constructive comments and suggestions to improve the quality of our manuscript "Mitigation of the double ITCZ syndrome in BCC-CSM2-MR through improving parameterizations of boundary-layer turbulence and shallow convection" by Yixiong Lu et al., submitted to *Geoscientific Model Development*.

We have revised our manuscript and answered all the comments given by the referee. Please find our detailed point-by-point responses to the comments below. The reviewer's comments are in black, and our responses are in red.

Best regards,

Yixiong Lu and all co-authors
* * *
**Response to Anonymous Referee #1**

Review of

**Mitigation of the double ITCZ syndrome in BCC-CSM2-MR through improving parameterizations of boundary-layer turbulence and shallow convection**
Yixiong Lu, Tongwen Wu, Yubin Li, & Ben Yang

**General**
The authors report the improvement in the known 'double ITCZ bias', common in most coupled climate models, in the BCC-CSM2-MR model. The authors aptly report the improvement caused by the implementation of new boundary layer and shallow cloud parameterisations. I am not an expert in the associated parameterizations, and so not in a position to critically assess the technical and physical aspects related to their implementation. I do have some comments with regards to the reported improvement in the double ITCZ bias and on the general presentation of the analysis. I find the work to be generally well presented and within the scope of GMD. With regards to the mitigation of the double ITCZ bias, I encourage the authors to take into consideration the comments provided below.

We would like to thank the reviewer for taking the time to carefully read our manuscript, for very valuable comments and suggestions and English grammatical corrections. We have revised our manuscript and answered all the comments given by the reviewer.

**Specific comments**
1. Adding the values of critical parameters used in the revised parameterization would help the readers and enable the reproducibility of the reported results (e.g., in Eq. 5).

Thank you for your suggestions. The parameter *A* in Eq. 5 is not a constant and the computation method has been added in the revised manuscript, as follows,

"Following Bretherton and Park (2009), *A* is expressed as

$$A = 0.1(1+30E), \tag{6}$$

where *E* is the evaporative enhancement, which is parameterized as

$$E = 0.8Lq_l^{ct} / \Delta s_{vl}. \tag{7}$$

*L* is the latent heat of vaporization, $q_l^{ct}$ is the cloud-top liquid water content, and $\Delta s_{vl}$ is the jump in the liquid virtual static energy across the cloud-top entrainment zone."

Please note that two more equations are included and the equations in the revised manuscript have been renumbered.

2. Line 27: the double ITCZ bias is seen year-round in the central and western Pacific, but only during the SH rainy season over the eastern Pacific (and Atlantic). (See Adam et al. 2018 and Li & Xie 2014)

Thank you for the comment. The double ITCZ bias indeed has distinct seasonal and regional characteristics, which have been clarified in the revised manuscript as follows,

"Specifically, the double ITCZ bias is primarily seen in the Pacific and Atlantic sectors, and during the southern hemisphere rainy season (Li and Xie, 2014; Adam et al., 2018)."

These two cited papers have been added in the reference list.

3. Lines 228-230: increased cloud fraction in the subtropical eastern Pacific cools surface waters which subduct and eventually end up in the cold tongue. It is worth mentioning this coupling mechanism here, which was shown to have an important effect by Burls et al. (2017).

Thank you for the comment. This coupling mechanism has been added in the revised manuscript, as follows,

"As shown in Burls et al. (2017), increased cloud fraction in the subtropical eastern Pacific has an important effect on the cold tongue by cooling sea surface waters which subduct and eventually end up in the equatorial Pacific."

The cited paper has been added in the reference list.

4.  Lines 255-256: both the surface temperature and surface temperature gradient have an important effect on boundary layer and deep convection, as shown by Back and Bretherton (2009). The effect of BL convergence by SST gradients is not accounted for in the analysis.

    Thank you for the comment. The boundary layer convergence is an important aspect of ITCZ. Thank you for reminding us to add this analysis. We added comparisons of surface convergence in Figure 10, replacing the wind stress magnitude. Discussions are included as follows,

    "Boundary layer convergence is primarily affected by SST gradients and can be usefully viewed as a forcing on deep convection over the tropical oceans (Back and Bretherton, 2009a, b). It is clearly shown in Figure 10 that NEW_cmip produces relative divergence in the southern Pacific between 5°S and 15°S compared to REF_cmip, which corresponds to the eliminated southern ITCZ rainfall band resulting from weaker deep convection."

[Figure]

Figure 10. The difference of annual mean wind stress vector and surface convergence (shaded, ×10$^{-6}$ s$^{-1}$) between NEW_cmip and REF_cmip.

The cited papers have been added in the reference list.

5.  Lines 272-274: Indeed, the biases in the eastern Pacific and Atlantic are reduced. However, a negative bias in the Equatorial Indian Ocean seems to get worse. Are A the biases in the Indian Ocean, as well as the changes in the revised model, also related to the BL parameterization?

    Thank you for the comment. Indeed, the simulated precipitation in the equatorial Indian Ocean has decreased in the revised model. Precipitation simulation is a

complex problem, involving many processes such as deep convection parameterization scheme and cloud microphysical parameterization scheme. The modification of boundary layer and shallow convection schemes in the model will affect the performance of deep convection and cloud microphysical schemes, and then cause changes in precipitation simulation. These discussions have been added in the revised manuscript, as follows,

"It should be noted that precipitation simulation is a complex problem, involving many processes such as deep convection and cloud microphysics. The modification of boundary layer and shallow convection schemes in the model will affect the performance of deep convection and cloud microphysical schemes, and then cause changes in precipitation simulation. For example, a negative bias in the equatorial Indian Ocean seems to get worse in NEW_cmip, which may be due to the indirect effects of changes in boundary layer and shallow convection parameterizations."

6. Fig. 4: The cold tongue bias seems to persist in the revised model. (This is mentioned lated in the analysis.) Since the cold tongue bias is known to be closely linked to the double ITCZ bias, it is interesting and worth highlighting that an improvement is achieved only in one aspect of the bias.

Thank you for the comment. Accurate simulation of the ITCZ-cold tongue complex depends on multiple atmospheric and oceanic processes, and this study focuses on the role of parameterized boundary-layer turbulence and shallow convection. Discussions are added in the revised manuscript as follows,

"It is interesting and worth highlighting that the cold tongue bias, which is closely linked to the double ITCZ bias, persists NEW_cmip, implying that other parameterized processes, e.g., deep convection and oceanic circulations, may play an important role in achieving more improvements."

7. Lines 286-293: Both the anti-symmetric and symmetric components of the precipitation bias are significant (e.g., Adam et al. 2016). I suspect that the equatorial precipitation index (Adam et al. 2016, 2018), which was found to be strongly correlated with other phenomena (Popp and Lutsko 2017), will be quite different from observations in the revised model, in particular since equatorial precipitation in Fig. 5 is lower than observed.

Thank you for the comment. We considered your suggestion as to computing the equatorial precipitation index. You are right. The equatorial precipitation index is not improved in the revised model. The observed equatorial precipitation indices are 0.136 in GPCP and 0.110 in CMAP, respectively. The simulated indices are much smaller, which is 0.013 in the original model and further reduces to -0.008 in the revised model. The worse index is consistent with less equatorial precipitation in the revised model. We have included these discussions in the revised manuscript,

as follows,

"On the other hand, the symmetric component of the tropical precipitation is quantified using the equatorial precipitation index $E_p$, defined as (Adam et al., 2016, 2018)

$$E_P = \frac{\overline{P}_{2°S-2°N}}{\overline{P}_{20°S-20°N}} - 1.$$ (10)

In the case of double ITCZ that straddle the equator and when the equatorial precipitation vanishes, $E_p$ assumes its minimum value $E_p = -1$. The more strongly peaked tropical precipitation is on the equator, the larger $E_p$. $E_p$ is also found to be largely correlated with the difference in zonal mean precipitation between the absolute maximum and the equator (Popp and Lutsko, 2017). The observed equatorial precipitation indices are 0.136 in GPCP and 0.110 in CMAP, respectively, whereas the simulated values are much smaller, which is 0.013 in REF_cmip and further reduces to −0.008 in NEW_cmip. The worse index in NEW_cmip is consistent with less equatorial precipitation shown in Figure 5."

Please note that a new equation is added and the equations in the revised manuscript have been renumbered. Also, the cited papers have been included in the reference list.

8.  Section 4.4 and Fig. 11: This is an odd Figure. According to classic theory, wouldn't it be the curl of the wind stress that affects the zonal ocean currents, rather than the intensity of the Walker circulation? In any case, Fig. 11 and the short treatment of this potentially important aspect seem perfunctory. I would suggest either omitting this section from the paper or providing a more detailed and complete analysis.

Thank you for your comments and suggestions. We have provided a more detailed and complete discussion about Figure 11. This figure is intended to illustrate the effects of enhanced southeasterly wind stress in and northwest of the Southeast Pacific region on the South Equatorial Current. We rewrote Section 4.4 as follows,

"Because of the strengthened southeasterly wind stress in and northwest of the SEP region, the south equatorial current in the upper ocean is enhanced. Figure 11 shows the longitude-depth cross section of zonal oceanic current and temperature averaged over 5°S – 10°S for the difference between NEW_cmip and REF_cmip. Compared with REF_cmip, the climatological westward zonal current in NEW_cmip over 5°S−10°S is enhanced by more than 8 cm/s above 120 m over the central to eastern Pacific. Further analysis indicates that the simulated subsurface temperature is reduced by more than 2 K above 80 m east of 135°W in NEW_cmip. Apparently, the enhanced westward ocean current over the whole zonal band helps transport cooler water from east to west and prevents the warm water in the western Pacific from extending eastward in NEW_cmip."

9. 5.2 Indeed the HR model seems to dramatically improve the representation of tropical precipitation. However the reader is left curious and confused. The authors claim that it is the UWMT that accounts fo the major improvement in the HR model but provide virtually no support for this claim.

Thank you for the comment. To support the claim that the major improvement in the HR model benefits from the UWMT boundary-layer turbulence and modified Hack shallow convection schemes, we have added a subplot in Figure 13 showing the precipitation simulation result from BCC-CSM2-HR with old boundary layer and shallow convection schemes. Correspondingly, we adjusted the sentences in the second paragraph of section 5.2, as follows,

"During the transition from BCC-CSM2-MR to BCC-CSM2-HR, the atmospheric component increased its horizontal resolution from T106 (~ 1.125°) to T266 (~ 0.45°) with a higher model top, and the physics package was essentially updated, especially the deep convection scheme. Furthermore, the oceanic component was upgraded to the Modular Ocean Model version 5 (MOM5). However, previous versions of BCC-CSM2-HR suffered from the double ITCZ syndrome until the UWMT and modified Hack schemes were introduced. Before improving parameterizations of boundary-layer turbulence and shallow convection, BCC-CSM2-HR simulated a southern rainfall band with excessive eastward extension over the central and eastern Pacific and two nearly parallel rain belts over the equatorial Atlantic (Figure 13a). This suggests that the boundary-layer and shallow convection schemes contribute primarily to the double ITCZ bias in BCC-CSM2-HR. The tropical precipitation patterns simulated in the frozen version of BCC-CSM2-HR, which is equipped with new boundary-layer turbulence and shallow convection schemes, barely manifest a double ITCZ, as shown in Figure 13b. The triangular-shaded dry region in the SEP reproduced by BCC-CSM2-HR resembles the observed much better than that simulated in the revised BCC-CSM2-MR, probably due to the improved interactions among the boundary-layer turbulence, shallow convection, and other processes. Anyway, improving parameterizations of boundary-layer turbulence and shallow convection shows robustness in mitigating the double ITCZ syndrome in different BCC coupled models."

[Figure]

[Figure]

Figure 13. Annual mean precipitation rate (mm day$^{-1}$) from (a) intermediate version of BCC-CSM2-HR with original boundary-layer turbulence and shallow convection schemes, and (b) frozen version of BCC-CSM2-HR with new boundary-layer turbulence and shallow convection schemes. The 3 mm day$^{-1}$ contour is included in bold for reference.

10. Lines 410-411: is this true also for the HR model?

Thank you for the question. In the high-resolution BCC-CSM2-HR, the cold tongue simulation seems to be unaffected by alleviating the double ITCZ bias, which may benefit from the improved deep convection parameterization. These discussions have been included in the revised manuscript.

**Technical/editorial comments**

14   promotes —> ameliorates

Done. We have changed 'promotes' to 'ameliorates'.

32   fake —> spurious

Done. Thank you for the correction.

37   impediment to what?

The sentence has been clarified as '… and it remains a serious impediment to model development'.

49   convection and cloud radiative effects

Done. 'convections and cloud' has been changed to 'convection and cloud radiative effects'. Thank you for the correction.

62   Previous attempts

Done. We have changed 'Previous studies' to 'Previous attempts'.

76   accounts for —> alleviates

Done. 'accounts for' is changed to 'alleviates'.

**References**
- Burls, N.J., Muir, L., Vincent, E.M. et al., 2017: Extra-tropical origin of equatorial Pacific cold bias in climate models with links to cloud albedo. Clim Dyn 49, 2093–2113
- Back, L.E. and C.S. Bretherton, 2009: A Simple Model of Climatological Rainfall and Vertical Motion Patterns over the Tropical Oceans. J. Climate, 22, 6477–6497
- Adam, O., T. Schneider, F. Brient, and T. Bischoff, 2016: Relation of the double-ITCZ bias to the atmospheric energy budget in climate models, Geophys. Res. Lett., 43, 7670–7677
- Adam, O., Schneider, T., and Brient, F., 2018: Regional and seasonal variations of the double-ITCZ bias in CMIP5 models, Clim. Dynam., 51, 101-117
- Popp, M., & Lutsko, N. J., 2017: Quantifying the zonal-mean structure of tropical precipitation. Geophysical Research Letters, 44, 9470–9478
- Li, G. and Xie S.-P., 2014: Tropical biases in CMIP5 multimodel ensemble: The

excessive equatorial Pacific cold tongue and double ITCZ problems. J. Climate, 27, 1765-1780

Thank you for providing these references which extend the breadth and depth of the manuscript. We have cited all the references.

---

## Referee Comment (RC2) · Anonymous Referee #2 · 14 Aug 2020

Review of Manuscript gmd-2020-40

Title: Mitigation of the double ITCZ syndrome in BCC-CSM2-MR through improving parameterizations of boundary-layer turbulence and shallow convection Authors: Yixiong Lu et al. Recommendation: major revision

Summary

The authors examine how the Pacific double ITCZ bias responds to modifying the boundary layer turbulence and shallow convection schemes in the BCC-CSM2-MR

[Figure]

GCM. They suggest than an improved representation of the stratocumulus-to-shallow-cumulus transition in the new parameterization leads to increased cloud cover and reduced SST in the southeastern tropical Pacific. This, they argue, alleviates the double ITCZ bias.

The paper is generally well written and concise. It is not clear, however, if the changes in the new model version objectively constitute an improvement. Rather, it seems that the modest improvement seen in the Pacific ITCZ is achieved at the expense of an unrealistically high cloud fraction and excessively cold SST in the southeastern tropical Pacific. This raises the question of the role of error compensation. I believe the results of the study are worth publishing but there needs to more objective/quantitative assessment of the bias reduction. There also needs to be more discussion regarding the aspects that deteriorate in the new model version, and discussion of the potential role of error compensation. Detailed comments follow below.

Major Comments

1) Figure 3 (longitude-height sections of cloud fraction) While REF_amip undeniably underestimates cloud fraction, NEW_amip certainly overestimates it, to the point where one wonders which version is better. Even qualitatively, the superiority of NEW_amip is not that obvious. In the Peruvian stratus region, e.g, there is a spurious offshore maximum at 95W, 850 hPa. Thus, it is important to have an objective measure of model performance. I suggest adding a table with pattern correlations and area-averaged root-mean-square errors (RMSEs) for all regions.

2) Figure 4 Again, it would be helpful to have an objective measure of improvements in the equatorial Pacific, like the RMSE. The unrealistically zonal orientation of the SPCZ seems to be pretty much the same in both experiments. It is true that the 3 mm/day contour does not extend to 90W anymore in NEW_amip, but that is just a very narrow protrusion whose elimination should have little impact on the area average. Interestingly, the improvements look more convincing in the equatorial Atlantic.

3) Figure 7 No mention is made of the cold bias in the target region that is incurred by using the new parameterization. Visual inspection suggests that the area-averaged RMSE of SST may actually deteriorate in NEW_cmip. Please calculate those metrics and discuss them.

4) Figure 10 How does the simulated wind stress compare to observations/reanalysis? Please add a panel.

5) Figure 11 I suggest removing this figure or expanding the analysis. While zonal advection is certainly a plausible mechanism for the cooling, a detailed heat budget analysis would be needed to make a convincing argument. Other processes, such as upwelling and vertical mixing may play an important role as well.

6) Figure 13 If this figure is to be kept there needs to be an additional panel showing performance before the introduction of the new schemes. Otherwise it is impossible to evaluate the improvement.

Minor Comments

1) ll. 32-33: Please mention some references for the Atlantic ITCZ bias (e.g. Richter al. 2014, Siongco et al. 2015).

2) ll. 59-60: Please provide some references for the claim that stratocumulus biases contribute to the double ITCZ problem. Also, some studies have found that shortwave radiation biases in marine stratocumulus regions are overcompensated for by excessive latent heat flux (even in AGCM-only simulations with prescribed observed SST), which suggests a different origin of the warm SST biases (de Szoeke and Xie 2008, Toniazzo and Woonough 2014, Vanniere et al. 2014, Xu et al. 2014, Zheng et al. 2011). This should be discussed.

3) ll. 69-70: Please provide a reference for this claim.

4) ll. 91-92: The atmospheric component ultimately traces its origins to the NCAR Community Atmospheric Model (CAM). It is important to note this origin and to explain

to what extent the BCC version has diverged over the years. Does the BCC model feature similar biases as current incarnations of CESM?

5) l. 143: What does "roots in level k+1" mean?

6) section 2.3, last para: In the light of the substantial progress made in the field, the LTS criterion appears crude and outdated. There must be more sophisticated criteria.

7) l. 256: "Below will clarity" -> "Below we examine"

8) l. 267: "triangular-shaded" -> "triangular" or "triangle-shaped"

9) Figure 6: Given the relatively small improvement in precipitation seen in Fig. 4, the large improvement in this figure is somewhat surprising. I guess the improvement is diluted in the annual mean (Fig. 4)?

10) ll. 411-412: "cold tough bias" -> "cold tongue bias"

11) The authors should discuss the work of Hourdin et al. (2020) as those authors also stress the importance of the marine boundary layer in tropical biases.

References

de Szoeke, S. P., and S. Xie, 2008: The Tropical Eastern Pacific Seasonal Cycle: Assessment of Errors and Mechanisms in IPCC AR4 Coupled Ocean–Atmosphere General Circulation Models. J. Climate, 21, 2573–2590, https://doi.org/10.1175/2007JCLI1975.1.

Hourdin, F., Rio, C., Jam, A., Traore, A.‐K., & Musat, I. (2020). Convective boundary layer control of the sea surface temperature in the tropics. Journal of Advances in Modeling Earth Systems, 12, e2019MS001988. https://doi.org/10.1029/2019MS001988

Richter, I., Xie, S., Behera, S.K. et al. Equatorial Atlantic variability and its relation to mean state biases in CMIP5. Clim Dyn 42, 171–188 (2014). https://doi.org/10.1007/s00382-012-1624-5

[Figure]

Siongco, A.C., Hohenegger, C. & Stevens, B. The Atlantic ITCZ bias in CMIP5 models. Clim Dyn 45, 1169–1180 (2015). https://doi.org/10.1007/s00382-014-2366-3

Toniazzo T, Woolnough S. Development of warm SST errors in the southern tropical Atlantic in CMIP5 decadal hindcasts. Clim Dyn 2014, 43:2889–2913.

Vannière B, Guilyardi E, Toniazzo T, Madec G, Woolnough S. A systematic approach to identify the sources of tropical SST errors in coupled models using the adjustment of initialised experiments. Clim Dyn 2014, 43:2261–2282.

Xu Z, Chang P, Richter I, Kim W, Tang G. Diagnosing southeast tropical Atlantic SST and ocean circulation biases in the CMIP5 ensemble. Clim Dyn 2014, 43:3123–3145.

Zheng Y, Shinoda T, Lin JL, Kiladis GN. Sea surface temperature biases under the stratus cloud deck in the Southeast Pacific Ocean in 19 IPCC AR4 coupled general circulation models. J Clim 2011, 24:4139–4164.
* * *

---

## Short Comment (SC2) · 22 Aug 2020

This is a great work to mitigate the double-ITCZ bias in BCC model by improving different model parameterizations. It was so impressive how much work have been done to achieve this. I have one comment related to the comparison between BCC-CSM2-HR and BCC-CSM2-MR on the double-ITCZ problem. As shown in my current work (Song and Zhang 2020), which shows that the increase of horizontal resolution of atmospheric model can reduce the seasonal double-ITCZ bias over the eastern Pacific by reducing the easterly wind bias crossing the Central America. We analyzed the CMIP5 models
by grouping them according to their model resolutions and designed experiments in CESM1. I always want to know to what extent this can also be applied to other models. For your case, I am wondering that if keeping the parameterizations same in both BCC-CSM2-HR and BCC-CSM2-MR, whether the higher-resolution model has smaller double-ITCZ bias. If you have done this kind of experiments, it is a great kindness of you to satisfy my curiosity. If you don't have such kinds of experiments handy, it is fine and just overlook my comments. Finally, this is a great work. I really love it.

Song, F., and G. Zhang, 2020: The impacts of horizontal resolution on the seasonallydependent biases of the northeastern Pacific ITCZ in coupled climate models, Journal of Climate, 33, 941-957.

---

## Author Comment (AC3) · 25 Aug 2020

Thank you for your kind comments. I have carefully read your paper, which shows the role of horizontal resolution in reducing the seasonal double-ITCZ bias over the eastern Pacific. Your excellent work provides more ideas for reducing the common double-ITCZ bias in coupled models.

As we shown in Figure 13 in our manuscript under review, the high-resolution BCC-CSM2-HR performs much better in tropical precipitation simulation with smaller double-

ITCZ bias. I guess higher resolution may contribute to the better results. However, with updated model physics and revised dynamics, BCC-CSM2-HR largely differs from BCC-CSM2-MR, and there is no chance to carry out experiments with same parameterizations in both of the two models.

On the other hand, we are developing next generation of BCC high-top models with two different resolutions but same physical schemes. Following your work, we will pay special attention to the impacts of horizontal resolution on the double-ITCZ bias during our subsequent model development. I'd like to keep in touch with you on this issue and to work in collaboration with you to examine whether higher resolution leads to smaller double-ITCZ bias in BCC models.

―――――――――――――――

---

## Author Comment (AC4) · 26 Aug 2020

Dear Referee #2,

We would like to thank you for your constructive comments and suggestions to improve the quality of our manuscript "Mitigation of the double ITCZ syndrome in BCC-CSM2-MR through improving parameterizations of boundary-layer turbulence and shallow convection" by Yixiong Lu et al., submitted to *Geoscientific Model Development*.

We have revised our manuscript and answered all the comments given by the referee. Please find our detailed point-by-point responses to the comments below. The reviewer's comments are in black, and our responses are in red.

Best regards,

Yixiong Lu and all co-authors
* * *
**Response to Anonymous Referee #2**

Review of Manuscript gmd-2020-40

**Title:** Mitigation of the double ITCZ syndrome in BCC-CSM2-MR through improving parameterizations of boundary-layer turbulence and shallow convection
Authors: Yixiong Lu et al.

**Recommendation:** major revision

**Summary**

The authors examine how the Pacific double ITCZ bias responds to modifying the boundary layer turbulence and shallow convection schemes in the BCC-CSM2-MR GCM. They suggest than an improved representation of the stratocumulus-to-shallow cumulus transition in the new parameterization leads to increased cloud cover and reduced SST in the southeastern tropical Pacific. This, they argue, alleviates the double ITCZ bias.

The paper is generally well written and concise. It is not clear, however, if the changes in the new model version objectively constitute an improvement. Rather, it seems that the modest improvement seen in the Pacific ITCZ is achieved at the expense of an unrealistically high cloud fraction and excessively cold SST in the southeastern tropical Pacific. This raises the question of the role of error compensation. I believe the results of the study are worth publishing but there needs to more objective/quantitative assessment of the bias reduction. There also needs to be more discussion regarding the aspects that deteriorate in the new model version, and discussion of the potential role of error compensation. Detailed comments follow

below.

We would like to thank the reviewer for taking the time to carefully read our manuscript, for very valuable comments and suggestions and English grammatical corrections. We have revised our manuscript and answered all the comments given by the reviewer. Following your suggestion, we have added two tables to present the quantitative assessment of the bias reduction. Moreover, discussions about the aspects that deteriorate in the modified model are also included in the revised manuscript. Please also note Figure 10 and 13 are expanded with more panels.

**Major Comments**

1) Figure 3 (longitude-height sections of cloud fraction) While REF_amip undeniably underestimates cloud fraction, NEW_amip certainly overestimates it, to the point where one wonders which version is better. Even qualitatively, the superiority of NEW_amip is not that obvious. In the Peruvian stratus region, e.g, there is a spurious offshore maximum at 95W, 850 hPa. Thus, it is important to have an objective measure of model performance. I suggest adding a table with pattern correlations and area-averaged root-mean-square errors (RMSEs) for all regions.

Thank you for the comment and suggestion. Figure 3 is intended to show a better representation of the qualitative characteristics of subtropical stratocumulus-to-cumulus transition. It is true that the vertical distribution of the cloud fraction needs further improvement. Following your suggestion, we have added a table to illustrate better model performance in NEW_amip and related discussion have been included in the revised manuscript, as follows,

"For more quantitative comparisons, Table 2 presents the area-averaged biases and root-mean-square errors (RMSEs) of the REF_amip and NEW_amip low cloud simulations to the GOCCP observations over the globe, in the tropics and for the five main subtropical marine stratocumulus regions shown in Figure 2. For all regions, REF_amip significantly underestimates the low cloud amounts and has large biases and RMSEs. Although the low cloud cover simulated by NEW_amip is still less, biases and RMSEs are substantially reduced for most regions, except for Canara where the cloud fraction is overestimated to some extent. Spatial pattern correlations are also calculated to evaluate the simulated low cloud distribution. For the global low cloud pattern, the correlation increases from 0.76 in REF_amip to 0.84 in NEW_amip. More obviously, the tropical pattern correlation increases from 0.72 in REF_amip to 0.89 in NEW_amip. Based on these objective measures, it is clear that NEW_amip performs better than REF_amip with improved parameterizations of BL turbulence and shallow convection."

Table 2. Evaluation of the low-level cloud fraction (%) from REF_amip and NEW_amip simulations against GOCCP observations. Shown are the area-averaged biases and root-mean-square errors (RMSEs) between simulated and observed low-level cloud amounts over the globe, in the tropics and for the five main subtropical marine stratocumulus regions, which is indicated in Figure 2. Pattern correlations are calculated for the global and tropical low-level cloud distribution in the simulations, respectively.

| Region | Bias | | RMSE | | Pattern Correlation | |
|---|---|---|---|---|---|---|
| | REF_amip | NEW_amip | REF_amip | NEW_amip | REF_amip | NEW_amip |
| Global | -12.48 | -8.35 | 12.57 | 8.49 | 0.76 | 0.84 |
| Tropical | -14.18 | -8.64 | 14.26 | 8.74 | 0.72 | 0.89 |
| Peruvian | -35.73 | -7.63 | 36.91 | 14.89 | | |
| Californian | -32.61 | -22.40 | 33.69 | 24.80 | | |
| Australian | -38.43 | -11.41 | 39.56 | 18.81 | | |
| Namibian | -28.37 | -3.45 | 30.11 | 12.55 | | |
| Canarian | -12.56 | 6.03 | 15.95 | 20.68 | | |

2) Figure 4 Again, it would be helpful to have an objective measure of improvements in the equatorial Pacific, like the RMSE. The unrealistically zonal orientation of the SPCZ seems to be pretty much the same in both experiments. It is true that the 3 mm/day contour does not extend to 90W anymore in NEW_amip, but that is just a very narrow protrusion whose elimination should have little impact on the area average. Interestingly, the improvements look more convincing in the equatorial Atlantic.

Thank you for the comment. Following your suggestion, we have calculated the area-averaged biases and RMSEs, and pattern correlations between simulated and observed precipitation rate in the tropical Pacific. Both biases and RMSEs significantly decrease in NEW_cmip, indicating that the simulation of the precipitation in the tropical Pacific is improved in NEW_cmip. The elimination of the narrow protrusion also leads to a slight increase in the pattern correlation. The manuscript has been revised as follows,

"Table 3 summarizes the area-averaged biases and RMSEs, and pattern correlations between simulated and observed precipitation rate in the tropical Pacific. Compared with GPCP (CMAP), the bias of simulated precipitation rate is reduced from 0.89 (0.33) in REF_cmip to 0.44 (-0.12) in NEW_cmip. Correspondingly, the RMSE decreases from 0.94 (0.48) in REF_cmip to 0.54 (0.36) in NEW_cmip. The elimination of excessive precipitation in the SEP leads to an increase of the pattern correlation, which is raised from 0.78 (0.80) in REF_cmip to 0.81 (0.81) in NEW_cmip. It is also interesting to note that the spurious southern precipitation belt in the equatorial Atlantic completely disappears in NEW_cmip, which agrees well with observations."

Table 3. Evaluation of the precipitation rate (mm day$^{-1}$) from REF_cmip and NEW_cmip simulations against GPCP and CMAP observational estimates. Shown are the area-averaged biases and root-mean-square errors (RMSEs), and pattern correlations between simulated and observed precipitation rate in the tropical Pacific (30°S−30°N, 120°E−90°W).

| Observational | Bias | | RMSE | | Pattern Correlation | |
|---|---|---|---|---|---|---|
| Data | REF_cmip | NEW_cmip | REF_cmip | NEW_cmip | REF_cmip | NEW_cmip |
| GPCP | 0.89 | 0.44 | 0.94 | 0.54 | 0.78 | 0.81 |
| CMAP | 0.33 | -0.12 | 0.48 | 0.36 | 0.80 | 0.81 |

3) Figure 7 No mention is made of the cold bias in the target region that is incurred by using the new parameterization. Visual inspection suggests that the area-averaged RMSE of SST may actually deteriorate in NEW_cmip. Please calculate those metrics and discuss them.

Thank you for the comment. We have mentioned the cold bias in the stratocumulus regions in NEW_cmip. The area-averaged RMSEs of SST have been calculated and indeed deteriorate in NEW_cmip. We have added discussion regarding this aspect that deteriorate in the modified model, as follows,

"It seems that the warm SST biases in REF_cmip are overcorrected in NEW_cmip by using new BL and shallow convection schemes, leading to a few degrees of cold bias in the SEP region. The area-averaged RMSE of SST in the tropical Pacific is 0.43 K in REF_cmip and actually deteriorates to 1.57 K in NEW_cmip. The common warm SST biases in CGCMs may come from several sources. Besides the underestimation of the shadowing effect due to a lack of stratocumulus that cover the SEP region, a poor representation of the oceanic surface cooling, by advection or mixing with the colder subsurface water, may also contribute to the warm biases (Richter, 2015). Also, some studies have found that shortwave radiation biases in marine stratocumulus regions are overcompensated for by excessive latent heat flux, which suggests a different origin of the warm SST biases (de Szoeke and Xie, 2008; Toniazzo and Woonough, 2014; Vanniere et al., 2014; Xu et al., 2014; Zheng et al., 2011). Recently, Hourdin et al. (2015) revealed that coupled models with warmer SST over the eastern tropical oceans present a lack of surface evaporative cooling in atmospheric simulations forced by SST. In the NEW_cmip simulation, an overestimation of the shadowing effect due to increased stratocumulus clouds may act to compensate for less surface evaporative cooling and make the sea surface cool enough to reduce precipitation in the SEP region."

4) Figure 10 How does the simulated wind stress compare to observations/reanalysis? Please add a panel.

Thank you for the question. We have added three panels for the results from reanalysis and two simulations. Please note that the wind stress magnitude is

replaced by surface convergence according to the comments of referee 1. Discussions are included in the revised manuscript, as follows,

"Figure 10 compares the annual mean surface wind stress vectors and surface convergence from REF_cmip and NEW_cmip simulations with JRA-55 reanalysis. In the eastern Pacific, the reanalysis shows convergence of northeasterly and southeasterly wind stresses in the northern ITCZ. The easterly and southeasterly wind stresses dominant central and eastern Pacific between 0° and 15°S, and no distinct convergence exists in these regions (Figure 10a). In the REF_cmip simulation (Figure 10b), the wind stress between 0° and 5°S is northeasterly compared to the observed easterlies, resulting in a convergence band in the central and eastern Pacific between 5°S and 10°S, which corresponding to the spurious southern ITCZ rainfall band. A prominent divergence zone also appears across the equatorial Pacific, which corresponds to the dry tongue in precipitation. The modified boundary-layer turbulence and shallow convection schemes result in increased southeasterly winds off the west coast of South America in NEW_cmip (Figure 10c). Specifically, the difference between NEW_cmip and REF_cmip clearly shows the strengthened southeasterly trade winds in the eastern Pacific between 5°S and 10°S (Figure 10d), corresponding to the stronger descending branch of the Walker circulation in NEW_cmip. Boundary layer convergence is primarily affected by SST gradients and can be usefully viewed as a forcing on deep convection over the tropical oceans (Back and Bretherton, 2009a, b). It is shown in Figure 10d that NEW_cmip produces relative divergence in the southern Pacific between 5°S and 15°S compared to REF_cmip, which corresponds to the eliminated southern ITCZ rainfall band resulting from weaker deep convection."

[Figure]

Figure 10. Annual mean wind stress vector and surface convergence (shaded, ×10$^{-6}$) from (a) JRA-55 reanalysis, (b) REF_cmip, (c) NEW_cmip, and (d) the difference between NEW_cmip and REF_cmip.

5) Figure 11 I suggest removing this figure or expanding the analysis. While zonal advection is certainly a plausible mechanism for the cooling, a detailed heat budget analysis would be needed to make a convincing argument. Other processes, such as upwelling and vertical mixing may play an important role as well.

Thank you for your comments and suggestions. Figure 11 is intended to illustrate the effects of enhanced southeasterly wind stress in and northwest of the Southeast Pacific region on the South Equatorial Current. We have provided a more detailed and complete discussion about this figure. We rewrote Section 4.4 as follows,

"Because of the strengthened southeasterly wind stress in and northwest of the SEP region, the south equatorial current in the upper ocean is enhanced. Figure 11 shows the longitude-depth cross section of zonal oceanic current and temperature averaged over 5°S – 10°S for the difference between NEW_cmip and REF_cmip. Compared with REF_cmip, the climatological westward zonal current in NEW_cmip over 5°S−10°S is enhanced by more than 8 cm/s above 120 m over the central to eastern Pacific. Further analysis indicates that the simulated subsurface temperature is reduced by more than 2 K above 80 m east of 135°W in NEW_cmip. Apparently, the enhanced westward ocean current over the whole zonal band helps transport cooler water from east to west and prevents the warm water in the western Pacific from extending eastward in NEW_cmip."

6) Figure 13 If this figure is to be kept there needs to be an additional panel showing performance before the introduction of the new schemes. Otherwise it is impossible to evaluate the improvement.

Thank you for the comment. To support the claim that the major improvement in the HR model benefits from the UWMT boundary-layer turbulence and modified Hack shallow convection schemes, we have added a subplot in Figure 13 showing the precipitation simulation result from BCC-CSM2-HR with old boundary layer and shallow convection schemes. Correspondingly, we adjusted the sentences in the second paragraph of section 5.2, as follows,

"During the transition from BCC-CSM2-MR to BCC-CSM2-HR, the atmospheric component increased its horizontal resolution from T106 (~ 1.125°) to T266 (~ 0.45°) with a higher model top, and the physics package was essentially updated, especially the deep convection scheme. Furthermore, the oceanic component was upgraded to the Modular Ocean Model version 5 (MOM5). However, previous versions of BCC-CSM2-HR suffered from the double ITCZ syndrome until the UWMT and modified Hack schemes were introduced. Before improving parameterizations of boundary-layer turbulence and shallow convection, BCC-CSM2-HR simulated a southern rainfall band with excessive eastward extension over the central and eastern Pacific and two nearly parallel rain belts over the

equatorial Atlantic (Figure 13a). This suggests that the boundary-layer and shallow convection schemes contribute primarily to the double ITCZ bias in BCC-CSM2-HR. The tropical precipitation patterns simulated in the frozen version of BCC-CSM2-HR, which is equipped with new boundary-layer turbulence and shallow convection schemes, barely manifest a double ITCZ, as shown in Figure 13b. The triangular-shaded dry region in the SEP reproduced by BCC-CSM2-HR resembles the observed much better than that simulated in the revised BCC-CSM2-MR, probably due to the improved interactions among the boundary-layer turbulence, shallow convection, and other processes. Anyway, improving parameterizations of boundary-layer turbulence and shallow convection shows robustness in mitigating the double ITCZ syndrome in different BCC coupled models."

[Figure]

Figure 13. Annual mean precipitation rate (mm day$^{-1}$) from (a) intermediate version of BCC-CSM2-HR with original boundary-layer turbulence and shallow convection schemes, and (b) frozen version of BCC-CSM2-HR with new boundary-layer turbulence and shallow convection schemes. The 3 mm day$^{-1}$ contour is included in bold for reference.

**Minor Comments**

1) ll. 32-33: Please mention some references for the Atlantic ITCZ bias (e.g. Richter et al. 2014, Siongco et al. 2015).

Thank you for the suggestion. We have cited these references and included them in the reference list in the revised manuscript.

2) ll. 59-60: Please provide some references for the claim that stratocumulus biases contribute to the double ITCZ problem. Also, some studies have found that shortwave radiation biases in marine stratocumulus regions are overcompensated for by excessive latent heat flux (even in AGCM-only simulations with prescribed observed SST), which suggests a different origin of the warm SST biases (de Szoeke and Xie 2008, Toniazzo and Woonough 2014, Vanniere et al. 2014, Xu et al. 2014, Zheng et al. 2011). This should be discussed.

Thank you for the comment. References have been added for the claim that stratocumulus biases contribute to the double ITCZ problem. We also included the discussion about the error compensation between shortwave radiation biases and latent heat biases. In particular, we cited the work of Hourdin et al. (2015) that identified a lack of surface evaporative cooling as a different origin of the warm SST biases. These discussions are added in section 4.2, as follows,

"The common warm SST biases in CGCMs may come from several sources. Besides the underestimation of the shadowing effect due to a lack of stratocumulus that cover the SEP region, a poor representation of the oceanic surface cooling, by advection or mixing with the colder subsurface water, may also contribute to the warm biases (Richter, 2015). Also, some studies have found that shortwave radiation biases in marine stratocumulus regions are overcompensated for by excessive latent heat flux, which suggests a different origin of the warm SST biases (de Szoeke and Xie, 2008; Toniazzo and Woonough, 2014; Vanniere et al., 2014; Xu et al., 2014; Zheng et al., 2011). Recently, Hourdin et al. (2015) revealed that coupled models with warmer SST over the eastern tropical oceans present a lack of surface evaporative cooling in atmospheric simulations forced by SST."

3) ll. 69-70: Please provide a reference for this claim.

Thank you for the suggestion. Two references are cited for the claim that the low-level cloud near the South American west coast is the steadiest and most persistent stratocumulus regime in the world, i.e.,
1. Wood, R. and Bretherton, C. S.: On the relationship between stratiform low cloud cover and lower-tropospheric stability, J. Climate, 19, 6425-6432, 2006.
2. Wood, R.: Stratocumulus clouds, Mon. Wea. Rev., 140, 2373-2423, 2012.

4) ll. 91-92: The atmospheric component ultimately traces its origins to the NCAR Community Atmospheric Model (CAM). It is important to note this origin and to explain to what extent the BCC version has diverged over the years. Does the

BCC model feature similar biases as current incarnations of CESM?

Thank you for your attention to the BCC model development. BCC-AGCM indeed originates from the CAM3 developed by NCAR. However, the dynamics in BCC-AGCM substantially different from the Eulerian spectral formulation of the dynamical equations in CAM3, and is featured by introducing a reference stratified atmospheric temperature and a reference surface pressure into the governing equations. Besides, new physical parameterizations have replaced the corresponding original ones, including a new convection scheme, a new cloud cover scheme, a dry adiabatic adjustment scheme, a modified scheme to calculate the air-sea turbulent fluxes, an empirical equation to compute the snow cover fraction, etc. The vertical discretization of the BCC-AGCM also differs from CAM3. Detailed model development description of the BCC-AGCM can be found in a series of relevant publications (Wu et al., 2008, 2010, 2012, 2013, 2019; Lu et al., 2013, 2020). So, BCC-AGCM has evolved into a largely different model and has different error characteristics from CAM. These descriptions have been added in the revised manuscript.

1. Wu Tongwen, Rucong Yu, and Fang Zhang, 2008: A modified dynamic framework for atmospheric spectral model and its application, J. Atmos.Sci., 65, 2235-2253.
2. Wu, T., Yu, R., Zhang, F., Wang, Z., Dong, M., Wang, L., Jin, X., Chen, D., Li, L.: The Beijing Climate Center atmospheric general circulation model: description and its performance for the present-day climate, Climate Dynamics, 34, 123-147, DOI 10.1007/s00382-008-0487-2, 2010.
3. Wu, T.: A mass-flux cumulus parameterization scheme for large-scale models: Description and test with observations, Clim. Dynam., 38, 725-744, doi:10.1007/s00382-011-0995-3, 2012.
4. Wu, T., Li, W., Ji, J., Xin, X., Li, L., Wang, Z, Zhang, Y., Li, J., Zhang, F., Wei, M., Shi, X., Wu, F., Zhang, L., Chu, M., Jie, W., Liu, Y., Wang, F., Liu, X., Li, Q., Dong, M., Liang, X., Gao, Y., Zhang, J.: Global carbon budgets simulated by the Beijing climate center climate system model for the last century. J Geophys Res Atmos, 118, 4326-4347. doi: 10.1002/jgrd.50320, 2013.
5. Wu, T., Lu, Y., Fang, Y., Xin, X., Li, L., Li, W., Jie, W., Zhang, J., Liu, Y., Zhang, L., Zhang, F., Zhang, Y., Wu, F., Li, J., Chu, M., Wang, Z., Shi, X., Liu, X., Wei, M., Huang, A., Zhang, Y., and Liu, X.: The Beijing Climate Center Climate System Model (BCC-CSM): the main progress from CMIP5 to CMIP6, Geosci. Model Dev., 12, 1573-1600, doi:10.5194/gmd-12-1573-2019, 2019.
6. Lu, Y., Zhou, M., and Wu, T.: Validation of parameterizations for the surface turbulent fluxes over sea ice with CHINARE 2010 and SHEBA data, Polar Res., 32, 20818, doi:10.3402/polar.v32i0.20818, 2013.
7. Lu, Y., Wu, T., Jie, W., Scaife, A. A., Andrews, M. B., and Richter, J. H.:

Variability of the stratospheric quasi-biennial oscillation and its wave forcing simulated in the Beijing Climate Center Atmospheric General Circulation Model, J. Atmos. Sci., 77, 149-165, doi:10.1175/JAS-D-19-0123.1, 2020.

5) l. 143: What does "roots in level k+1" mean?

Thank you for the question. We have changed "roots in level k+1" to "originated from level k+1".

6) section 2.3, last para: In the light of the substantial progress made in the field, the LTS criterion appears crude and outdated. There must be more sophisticated criteria.

Thank you for the comment. The LTS criterion is relatively crude and we also notice that there are some improved criteria. Testing more sophisticated criteria is in our future study plans. Discussion about this aspect is included in the revised manuscript, as follows,

"It should be noted that the LTS criterion has been developed into physically more plausible formula. Wood and Bretherton (2006) modified the LTS to account for the strength of the BL inversion, called the estimated inversion strength (EIS) which is shown to be more useful than LTS for determining low cloud cover in the present climate. EIS is then further revised to take into account cloud-top entrainment and transformed into the estimated cloud-top entrainment index (ECTEI), which shows dependence on sea surface temperature (Kawai et al., 2017). Impacts of more sophisticated criteria on cloud representation and precipitation simulation in BCC-CSM2-MR is beyond the scope of this paper and will be explored in future work."

1. Wood, R. and Bretherton, C. S.: On the relationship between stratiform low cloud cover and lower-tropospheric stability, J. Climate, 19, 6425-6432, 2006.
2. Kawai, H., Koshiro, T., Webb, M. J.: Interpretation of factors controlling low cloud cover and low cloud feedback using a unified predictive index, J. Climate, 30, 9119-9131, 2017.

7) l. 256: "Below will clarity" −> "Below we examine"

Revised. Thank you for the correction.

8) l. 267: "triangular-shaded" −> "triangular" or "triangle-shaped"

Done. Thank you for the correction. We have corrected "triangular-shaded" to "triangular-shaped".

9) Figure 6: Given the relatively small improvement in precipitation seen in Fig. 4, the large improvement in this figure is somewhat surprising. I guess the improvement is diluted in the annual mean (Fig. 4)?

You are right. In fact, the double-ITCZ bias presents obvious seasonal variations. In the BCC model, this bias is most prominent in the cold season, e.g., from January to April. If we look at the annual average, the improvement is weaker.

10) ll. 411-412: "cold tough bias" −> "cold tongue bias"

Revised. Thank you for the correction.

11) The authors should discuss the work of Hourdin et al. (2020) as those authors also stress the importance of the marine boundary layer in tropical biases.

We have carefully read this important paper, which claims that the surface evaporative cooling plays a role as large as the shadowing effect of stratocumulus. We will pay special attention to this control mechanism in our future model development. Actually, our study follows the eddy diffusion mass flux (EDMF) approach, aiming to unify BL and shallow convective processes as in the IPSL model. More work should be done to improve the BL convection represented by the modified Hack scheme used in this study. We have added a paragraph in the section of Summary and conclusions to discuss the implication of Hourdin et al. (2020) and its inspiration for our future work, as follows,

"The BL processes can not only affect SST by changing the stratocumulus and its radiative effect, but also control the surface evaporative cooling by convective transport of humidity at the surface and then SST (Hourdin et al., 2020). Using a mass flux representation of the organized structures of the convective BL coupled to eddy diffusion, Hourdin et al. (2020) showed that an increased near-surface drying led to a reduction of the warm bias in the eastern tropical oceans in the Institute Pierre Simon Laplace coupled model, IPSL-CM6A. They concluded that a good representation of BL convection is required to maintain a strong contrast between trade winds cumulus regions and stratocumulus regions. Similarly, this study adopts the eddy diffusion mass flux (EDMF) approach, which seeks to unify BL and shallow convective processes by the marriage of UWMT and modified Hack schemes. However, there are still large discrepancies in the simulated Sc-to-Cu transition compared to observations, as shown in Figure 3, which suggests that parameterization of BL convection should be further improved. Moreover, the role of surface evaporative cooling needs to be explored when improving representation of BL convection."

**References**

de Szoeke, S. P., and S. Xie, 2008: The Tropical Eastern Pacific Seasonal Cycle:

Assessment of Errors and Mechanisms in IPCC AR4 Coupled Ocean–Atmosphere General Circulation Models. J. Climate, 21, 2573–2590, https://doi.org/10.1175/2007JCLI1975.1.

Hourdin, F., Rio, C., Jam, A., Traore, A.ă˘ARˇ K., & Musat, I. (2020). Convective boundary layer control of the sea surface temperature in the tropics. Journal of Advances in Modeling Earth Systems, 12, e2019MS001988. https://doi.org/10.1029/2019MS001988

Richter, I., Xie, S., Behera, S.K. et al. Equatorial Atlantic variability and its relation to mean state biases in CMIP5. Clim Dyn 42, 171–188 (2014). https://doi.org/10.1007/s00382-012-1624-5

Siongco, A.C., Hohenegger, C. & Stevens, B. The Atlantic ITCZ bias in CMIP5 models. Clim Dyn 45, 1169–1180 (2015). https://doi.org/10.1007/s00382-014-2366-3

Toniazzo T, Woolnough S. Development of warm SST errors in the southern tropical Atlantic in CMIP5 decadal hindcasts. Clim Dyn 2014, 43:2889–2913.

Vannière B, Guilyardi E, Toniazzo T, Madec G, Woolnough S. A systematic approach to identify the sources of tropical SST errors in coupled models using the adjustment of initialised experiments. Clim Dyn 2014, 43:2261–2282.

Xu Z, Chang P, Richter I, Kim W, Tang G. Diagnosing southeast tropical Atlantic SST and ocean circulation biases in the CMIP5 ensemble. Clim Dyn 2014, 43:3123–3145.

Zheng Y, Shinoda T, Lin JL, Kiladis GN. Sea surface temperature biases under the stratus cloud deck in the Southeast Pacific Ocean in 19 IPCC AR4 coupled general circulation models. J Clim 2011, 24:4139–4164.

Thank you for providing these references which extend the breadth and depth of the manuscript. We have cited all the references.

---

## Referee Report (RR1)

Review comments

**Mitigation of the double ITCZ syndrome in BCC-CSM2-MR through improving parameterizations of boundary-layer turbulence and shallow convection**

Yixiong Lu, Tongwen Wu, Yubin Li, Ben Yang

This study assesses the impact of an improved parameterization of turbulence and shallow convection on the precipitation bias in the ITCZ region. According to the manuscript, the precipitation bias is reduced through an improvement in the transition from stratocumulus to shallow cumulus convection over eastern subtropical oceans. The increased low-level cloud fraction reduces net surface shortwave radiation in the southeastern Pacific, which induces stronger and wider subsiding motion of the Walker circulation and remotely impacts the precipitation band in the ITCZ.

The study is well motivated, the manuscript is clearly written and the topic is interesting and highly relevant. It is evident that turbulence and shallow convection cause a substantial impact on the ITCZ precipitation bias and the study has impressive results. However, several weak points have to be addressed to make the conclusions convincing and acceptable for publication.

**Major points:**

1) Although the bias in precipitation rate is reduced, the double ITCZ pattern is not significantly changed in the NEW_cmip experiment. This, in my opinion, is a weak point of the study and cannot support the main conclusion that the double ITCZ band is mitigated. Moreover, the result presented in Fig. 13 shows that it is necessary to increase the model resolution to achieve the improvement in the ITCZ pattern, which signifies that other processes or interactions between the processes are more (or at least equally) relevant than the improvements in the representation of turbulence and shallow convection and the Sc to Cu transition.

2) The manuscript demonstrates a significant improvement in the double ITCZ precipitation band in the result only marginally shown in Fig. 13 where the HR model is used. This is a great result, however, it is not well explained. I suspect that the main cause of the double ITCZ syndrome lies in the difference between the performance of BCC-CSM2-MR and BCC-CSM2-HR when the turbulence and shallow convection schemes are improved in both model configurations.

3) It is not demonstrated how the improved Sc to Cu transition is contributing to the reduction of the precipitation bias. Furthermore, it is not clear why the REF_amip and NEW_amip simulations are used to demonstrate the improvement in the Sc to Cu transition instead of evaluating these processes directly in the REF_cmip and NEW_cmip simulations. The ITCZ is evaluated in the REF and NEW_cmip simulations, so it is expected that the changes in the clouds and the Sc to Cu transition are also investigated in the REF and NEW_cmip simulations.

4) The transition from Sc to Cu is improved in NEW_amip, however, the cloud amount is largely overestimated. This might be leading to changes in the precipitation rates and a decrease of the bias, however, it might just be a spurious compensating effect of the overestimated cloud amount.

5) It cannot be excluded that the improved turbulence and shallow convection schemes act locally to reduce the bias in precipitation in the ITCZ region. Especially because the ITCZ spatial pattern does not differ significantly between REF and NEW_cmip simulations, which would assumably be expected if the large scale circulation is changed. BCC-CSM2-HR thus should also be included in the analysis as one of the main experiments to assess this important question.

Other detailed comments:

**Fig. 1:** Here the differences between NEW_amip and REF_amip are shown instead of NEW_amip - CERES-EBAF. This shows the performance of NEW_amip relative to REF_amip, but no evaluation of NEW_amip simulation. Later on, it is shown that the cloud amounts are overestimated in NEW_amip, so this information is missing here - how much are the cloud effects overestimated in NEW_amip?

**Section 3.3** does not discuss an overestimated cloud amount in NEW_amip. Please add such a discussion.

**Figs 1 and 3:** How does the overestimated cloud amount in NEW_amip affect the main conclusion of the study? If the cloud effect is exaggerated, its impact on the ITCZ might also not be realistic.

**L245:** The bias is shown in Figure 1c, please refer to this figure here: „weak bias in the magnitude of TOA SWCRF over these regions in REF_amip „

 - It is not well noted when the analysis of the results switches from _amip to _cmip simulations. Are there any qualitative or quantitative differences in the impact of the turbulence and convection schemes on the ITCZ bias in _cmip compared to _amip? If it is necessary to show the analysis of the _amip simulations, please explain why the analysis switches between these two configurations.

- It is confusing that low-level clouds are validated only in the _amip experiments but ITCZ bias is not discussed for these experiments, while clouds are not evaluated in _cmip but the ITCZ bias is discussed only for these simulations. How are the clouds changed in NEW_cmip compared to REF_cmip? Also, are there any changes in the ITCZ precipitation band in NEW_amip compared to REF_amip?

**Line 305**: the improvement is not so visible from these plots. I would argue that there is a quantitative difference in the precipitation rates, but no qualitative improvement in the ITCZ precipitation bands. I would suggest a more detailed plot with differences (biases). The difference in pattern correlations proves this point because the change from 0.78 to 0.81 (GPCP) or from 0.80 to 0.81 (CMAP) is not very notable.

**Line 355:** It is not well explained why the excessive precipitation south of the equator in boreal winter and spring is reduced and closer to the observation in NEW_cmip. How is this connected to Sc-Cu transition in subtropical regions?

**Line 368.** The annual mean SST in NEW_cmip simulation is not presented, only its difference to the REF_cmip is presented, which makes it very difficult to compare between the experiments and assess the changes and impacts of clouds.

**Line 467:** The change in the pattern of ITCZ in BCC-CSM2-HR due to the improved turbulence and shallow convection scheme is a significant and very interesting result. Figure 13 is more convincing than previous figures that were based on the coarser-resolution model. However, the causes of this improvement are not explained in the present manuscript. This marginal result is, in my opinion, more relevant and could explain the ITCZ bias better than the main experiments of the study.

**Line 494**: „Better consistency between the BL turbulence scheme and the shallow convection scheme results in better simulation of the Sc-to-Cu transition.“ This is not shown in the NEW_cmip simulation, so there is no evidence that the Sc-to-Cu transition is improved in the _cmip simulations. There is no guarantee that clouds behave the same in AMIP and CMIP simulations.

---

## Author Response (AR2)

Dear Referee #3,

We appreciate the opportunity to modify our paper according to the comments of the reviewer for the manuscript entitled "Mitigation of the double ITCZ syndrome in BCC-CSM2-MR through improving parameterizations of boundary-layer turbulence and shallow convection" by Yixiong Lu et al., submitted to *Geoscientific Model Development*.

We would like to thank you for your constructive comments and suggestions to improve the quality of our manuscript. We have revised our manuscript and answered all the comments given by the referee. Please find our detailed point-by-point responses to the comments below. The reviewer's comments are in black, and our responses are in red. A tracked-changes version of the revised manuscript is also uploaded for your reference.

Please note that one new figure is added and two figures are modified in the revised manuscript. Correspondingly, the figures in the revised manuscript have been renumbered.

Best regards,

Yixiong Lu and all co-authors
* * *
**Response to Anonymous Referee #3**

Review comments

**Mitigation of the double ITCZ syndrome in BCC-CSM2-MR through improving parameterizations of boundary-layer turbulence and shallow convection**

Yixiong Lu, Tongwen Wu, Yubin Li, Ben Yang

This study assesses the impact of an improved parameterization of turbulence and shallow convection on the precipitation bias in the ITCZ region. According to the manuscript, the precipitation bias is reduced through an improvement in the transition from stratocumulus to shallow cumulus convection over eastern subtropical oceans. The increased low-level cloud fraction reduces net surface shortwave radiation in the southeastern Pacific, which induces stronger and wider subsiding motion of the Walker circulation and remotely impacts the precipitation band in the ITCZ.

The study is well motivated, the manuscript is clearly written and the topic is interesting and highly relevant. It is evident that turbulence and shallow convection cause a substantial impact on the ITCZ precipitation bias and the study has impressive results. However, several weak points have to be addressed to make the conclusions convincing

and acceptable for publication.

We would like to thank the reviewer for the valuable comments and suggestions. We have carefully answered all the comments given by the referee.

**Major points:**

1) Although the bias in precipitation rate is reduced, the double ITCZ pattern is not significantly changed in the NEW_cmip experiment. This, in my opinion, is a weak point of the study and cannot support the main conclusion that the double ITCZ band is mitigated. Moreover, the result presented in Fig. 13 shows that it is necessary to increase the model resolution to achieve the improvement in the ITCZ pattern, which signifies that other processes or interactions between the processes are more (or at least equally) relevant than the improvements in the representation of turbulence and shallow convection and the Sc to Cu transition.

Thank you for the comment. The double ITCZ syndrome in coupled GCMs is characterized by excessive precipitation in the southeastern Pacific and a spurious southern rain band in the tropical Atlantic basin, which are mainly presented in cold seasons. It can be seen in Figure 7 in the revised manuscript that the excessive precipitation is reduced in NEW_cmip and closer to the observation. Besides, the asymmetry of the tropical precipitation is improved, with the asymmetry index increases from -0.024 in REF_cmip to 0.147 in NEW_cmip, which is much closer to the observed 0.194. From this point of view, we say that the double ITCZ syndrome is mitigated in the revised BCC-CSM2-MR.

We agree that the alleviation of the double ITCZ in BCC-CSM2-HR due to the improved turbulence and shallow convection scheme is more significant. In the manuscript, we state that BCC-CSM2-HR resembles the observed much better than the revised BCC-CSM2-MR, probably due to the improved interactions among the boundary-layer turbulence, shallow convection, and other processes such as deep convection in a higher resolution. We are also very interested in the results of BCC-CSM2-HR and will prepared a separate paper to discuss the causes of the improvement in detail.

2) The manuscript demonstrates a significant improvement in the double ITCZ precipitation band in the result only marginally shown in Fig. 13 where the HR model is used. This is a great result, however, it is not well explained. I suspect that the main cause of the double ITCZ syndrome lies in the difference between the performance of BCC-CSM2-MR and BCC-CSM2-HR when the turbulence and shallow convection schemes are improved in both model configurations.

Thank you for the comment. As mentioned above, we agree that the alleviation of the double ITCZ in BCC-CSM2-HR due to the improved turbulence and shallow

convection scheme is more significant. BCC-CSM2-HR is very different from BCC-CSM2-MR, not only has higher resolution, but also has an updated physics package. In this manuscript, we show the results from BCC-CSM2-HR to demonstrate the robustness of the alleviated double ITCZ due to improvements in parameterizations of boundary-layer turbulence and shallow convections. We also agree that BCC-CSM2-HR resembles the observed much better than the revised BCC-CSM2-MR, probably due to the difference between the performance of BCC-CSM2-MR and BCC-CSM2-HR when the turbulence and shallow convection schemes are improved in both model configurations. This needs careful analysis and we will prepare a separate paper to discuss the causes of the improvement in BCC-CSM-HR in detail. Thank you for your constructive suggestion.

3) It is not demonstrated how the improved Sc to Cu transition is contributing to the reduction of the precipitation bias. Furthermore, it is not clear why the REF_amip and NEW_amip simulations are used to demonstrate the improvement in the Sc to Cu transition instead of evaluating these processes directly in the REF_cmip and NEW_cmip simulations. The ITCZ is evaluated in the REF and NEW_cmip simulations, so it is expected that the changes in the clouds and the Sc to Cu transition are also investigated in the REF and NEW_cmip simulations.

Thank you for the comment. The negative cloud amount biases off the west coast of South America, which are common in CGCMs, are regarded as a major cause for the double ITCZ problem. It is believed that the underestimated cloud cover leads to more net heat flux into the ocean and the warm SST biases in the SEP, which is associated with stronger convections and precipitation. A prominent feature of the low-level cloud in the SEP is that the stratocumulus regime progressively transforms into the trade cumulus regime moving downstream off the coast. In the transition, interactions between BL turbulence and shallow convection play a critical role. In most CGCMs, the stratocumulus is underestimated. We have added some discussions about this point at the end of section 4.2 in the revised manuscript, as follows,

"In the SEP, where the Sc-to-Cu transition is prominent, the improved cloud simulation in NEW_cmip will enhance the shadowing effect due to increased stratocumulus, leading to oceanic surface cooling as seen in Figure 8f. According to the relationship between precipitation and SST, cooled SST cause reduced precipitation. This can explain why the excessive precipitation in the SEP is reduced and closer to the observation in NEW_cmip."

It should be noted that the double ITCZ bias is basically a common bias in coupled ocean-atmosphere models. In atmosphere-only experiments, there is almost no such problem. Changes in cloud and its radiative effect can induce changes in SST, which will give feedbacks to atmospheric boundary-layer processes and convections. To test the response of low-cloud simulation to the parameterizations of boundary-layer turbulence and shallow convection, we prescribed the SST and conducted atmosphereonly experiments to prevent changes of simulated clouds from SST feedbacks. Then, we performed coupled experiments to see the response of the precipitation simulation including SST feedbacks.

4) The transition from Sc to Cu is improved in NEW_amip, however, the cloud amount is largely overestimated. This might be leading to changes in the precipitation rates and a decrease of the bias, however, it might just be a spurious compensating effect of the overestimated cloud amount.

Thank you for the comment. There are indeed doubts in this regard. So more investigations on the role of boundary-layer turbulence and shallow convection schemes in the double ITCZ formation in different CGCMs are desired. For simplicity, the UWMT and modified Hack schemes are employed in the high-resolution BCC-CSM2-HR, which is largely different from BCC-CSM2-MR with respect to model physics and dynamics, to examine the robustness of the alleviated double ITCZ through improving parameterizations of boundary-layer turbulence and shallow convection. As a result, the tropical precipitation patterns simulated in BCC-CSM2-HR, which is equipped with new boundary-layer turbulence and shallow convection schemes, barely manifest a double ITCZ. The results from BCC-CSM2-HR demonstrate the reliability of applying new boundary-layer turbulence and shallow convection schemes in alleviating the double ITCZ bias.

5) It cannot be excluded that the improved turbulence and shallow convection schemes act locally to reduce the bias in precipitation in the ITCZ region. Especially because the ITCZ spatial pattern does not differ significantly between REF and NEW_cmip simulations, which would assumably be expected if the large scale circulation is changed. BCC-CSM2-HR thus should also be included in the analysis as one of the main experiments to assess this important question.

Thank you for the suggestion. The question you mentioned above is important. We would like to prepare a separate paper to discuss the causes of the improvement in BCC-CSM-HR in detail.

Other detailed comments:

**Fig. 1**: Here the differences between NEW_amip and REF_amip are shown instead of NEW_amip - CERES-EBAF. This shows the performance of NEW_amip relative to REF_amip, but no evaluation of NEW_amip simulation. Later on, it is shown that the cloud amounts are overestimated in NEW_amip, so this information is missing here - how much are the cloud effects overestimated in NEW_amip?

Thank you for the comment. We have added two panels in Figure 1 to show the TOA SWCRF differences between NEW_amip and CERES-EBAF (Figure 1e), and also the annual climatology from NEW_amip (Figure 1c). Correspondingly, we discussed the

evaluation of NEW_amip simulation in section 3.1, as follows,

"With new configurations of BL turbulence and shallow convection schemes, NEW_amip presents a global mean TOA SWCRF of −54.45 W m$^{-2}$ (Figure 1c). Compared with REF_amip (Figure 1f), NEW_amip shows a considerably increased magnitude of SWCRF over the eastern subtropical ocean regions, suggesting that the representation of low-level marine stratocumulus is enhanced. However, the SWCRF is overestimated in NEW_amip over these regions (Figure 1e)."

[Figure]

Figure 1. Annual mean climatologies of TOA SWCRF (W m$^{-2}$) from (a) CERES-EBAF observations (from 2001 to 2010), (b) REF_amip, (c) NEW_amip, and the differences between (d) REF_amip and observations, (e) NEW_amip and observations, (f) NEW_amip and REF_amip. Shown atop panels (a), (b) and (c) are global mean values.

**Section 3.3** does not discuss an overestimated cloud amount in NEW_amip. Please add such a discussion.

Thank you for the comment. We have added a discussion about the overestimated cloud amount in NEW_amip in section 3.3, as follows,

"A possible reason for the over-extension of Sc in NEW_amip may be that the decoupling criterion added to the Hack shallow convection scheme is too strong, leading to the weak vertical mixing over the eastern edge of the shallow cumulus regime. The weak vertical mixing may further suppress the upward transport of water vapor, resulting in excessive low-level cloud amounts."

**Figs 1 and 3**: How does the overestimated cloud amount in NEW_amip affect the main conclusion of the study? If the cloud effect is exaggerated, its impact on the ITCZ might also not be realistic.

Thank you for the question. The double ITCZ is a prominent bias in the coupled atmosphere-ocean GCMs, which may result from discrepancies in the representation of both atmospheric and oceanic processes. The main conclusion of this study is that improving parameterizations of BL turbulence and shallow convection is an effective way to reduce the double ITCZ in CGCMs. The exaggerated cloud amount and cloud effect in modified BCC-CSM2-MR can be explained as a compensation for related biases in other physical processes. We have added these discussions in the section of Summary and conclusions.

**L245**: The bias is shown in Figure 1c, please refer to this figure here: "weak bias in the magnitude of TOA SWCRF over these regions in REF_amip"

Done. We have referred to this figure here.

- It is not well noted when the analysis of the results switches from _amip to _cmip simulations. Are there any qualitative or quantitative differences in the impact of the turbulence and convection schemes on the ITCZ bias in _cmip compared to _amip? If it is necessary to show the analysis of the _amip simulations, please explain why the analysis switches between these two configurations.

Thank you for the comment. Changes of cloud and its radiative effect can induce changes in SST, which will give feedbacks to atmospheric boundary-layer processes and convections. To test the response of low-cloud simulation to the parameterizations of boundary-layer turbulence and shallow convection, we prescribed the SST and conducted atmosphere-only experiments to prevent changes of simulated clouds from SST feedbacks. Then, we performed coupled experiments to see the response of the precipitation simulation including SST feedbacks.

- It is confusing that low-level clouds are validated only in the _amip experiments but ITCZ bias is not discussed for these experiments, while clouds are not evaluated in _cmip but the ITCZ bias is discussed only for these simulations. How are the clouds changed in NEW_cmip compared to REF_cmip? Also, are there any changes in the ITCZ precipitation band in NEW_amip compared to REF_amip?

Thank you for the question. Actually, the double ITCZ bias is basically a common bias in coupled ocean-atmosphere models. In atmosphere-only experiments, there is almost no such problem. So we ITCZ bias is not discussed for _amip experiments. The precipitation simulated in REF_amip and NEW_amip are shown below.

[Figure]

Annual mean precipitation rate (mm day$^{-1}$) from (a) GPCP, (b) CMAP, (c) REF_amip, and (d) NEW_amip. The 3 mm day$^{-1}$ contour is included in bold for reference.

**Line 305**: the improvement is not so visible from these plots. I would argue that there is a quantitative difference in the precipitation rates, but no qualitative improvement in the ITCZ precipitation bands. I would suggest a more detailed plot with differences (biases). The difference in pattern correlations proves this point because the change from 0.78 to 0.81 (GPCP) or from 0.80 to 0.81 (CMAP) is not very notable.

Thank you for the comment. Following your suggestion, we have added a figure to show the differences of annual mean precipitation rate between the NEW_cmip and REF_cmip experiments. Discussion about this figure is included in the revised manuscript, as follows,

"A more detailed plot with differences of annual-mean precipitation rates between NEW_cmip and REF_cmip is shown in Figure 5. The differences are mainly presented in the tropics, with decreased precipitation in the western Pacific, the SPCZ, the equatorial Atlantic oceans over 0−10°S, and western Indian ocean. Particularly, the precipitation rates over the SPCZ and the equatorial southern Atlantic are reduced up to 4 mm day$^{-1}$, leading to an alleviation of the double ITCZ bias in the NEW_cmip

experiment."

[Figure]

Figure 5. Differences of annual-mean precipitation rate (mm day$^{-1}$) between NEW_cmip and REF_cmip.

**Line 355**: It is not well explained why the excessive precipitation south of the equator in boreal winter and spring is reduced and closer to the observation in NEW_cmip. How is this connected to Sc-Cu transition in subtropical regions?

Thank you for the comment. We have included some discussions about this point at the end of section 4.2 in the revised manuscript, as follows,

"In the SEP, where the Sc-to-Cu transition is prominent, the improved cloud simulation in NEW_cmip will enhance the shadowing effect due to increased stratocumulus, leading to oceanic surface cooling as seen in Figure 8f. According to the relationship between precipitation and SST, cooled SST cause reduced precipitation. This can explain why the excessive precipitation in the SEP is reduced and closer to the observation in NEW_cmip."

**Line 368**. The annual mean SST in NEW_cmip simulation is not presented, only its difference to the REF_cmip is presented, which makes it very difficult to compare between the experiments and assess the changes and impacts of clouds.

Thank you for the comment. We have modified the Figure 8 in the revised manuscript, to show the annual mean SST in NEW_cmip and its bias compared with observations. The corresponding discussion is included in the manuscript, as follows,

"When the new boundary-layer turbulence and shallow convection schemes are used, the simulated sea surface water is cooled down almost in the entire Pacific, especially in the SEP where the warm water in REF_cmip is cooled down by up to 4 K (Figure 8f)"

[Figure]

Figure 8. Annual mean sea surface temperature (°C) from (a) HadISST, (b) REF_cmip, and (c) NEW_cmip, and the difference between (d) REF_cmip and HadISST, (e) NEW_cmip and HadISST, and (f) NEW_cmip and REF_cmip.

**Line 467**: The change in the pattern of ITCZ in BCC-CSM2-HR due to the improved turbulence and shallow convection scheme is a significant and very interesting result. Figure 13 is more convincing than previous figures that were based on the coarser-resolution model. However, the causes of this improvement are not explained in the present manuscript. This marginal result is, in my opinion, more relevant and could explain the ITCZ bias better than the main experiments of the study.

Thank you for the comment. You are right. The change in the pattern of ITCZ in BCC-CSM2-HR due to the improved turbulence and shallow convection scheme is more significant. Actually, the new scheme is developed and tested first in BCC-CSM2-MR, just after the outputs from BCC-CSM2-MR were provided to CMIP6 and found to suffer from the double ITCZ bias. At that time, BCC-CSM2-HR is still in its embryonic form. So we wrote most of the manuscript based on BCC-CSM2-MR. In the development of BCC-CSM2-HR, not only the dynamic framework was modified, but also the physics packages, especially the deep convection scheme. In the manuscript, we state that BCC-CSM2-HR resembles the observed much better than the revised

BCC-CSM2-MR, probably due to the improved interactions among the boundary-layer turbulence, shallow convection, and other processes such as deep convection in a higher resolution. We are also very interested in the results of BCC-CSM2-HR and will prepared a separate paper to discuss the causes of the improvement in detail.

**Line 494**: "Better consistency between the BL turbulence scheme and the shallow convection scheme results in better simulation of the Sc-to-Cu transition." This is not shown in the NEW_cmip simulation, so there is no evidence that the Sc-to-Cu transition is improved in the _cmip simulations. There is no guarantee that clouds behave the same in AMIP and CMIP simulations.

[revised manuscript text omitted]

---

## Author Response (AR3)

Dear Editor,

We would like to thank you for your valuable comments for the manuscript entitled "Mitigation of the double ITCZ syndrome in BCC-CSM2-MR through improving parameterizations of boundary-layer turbulence and shallow convection" by Yixiong Lu et al., submitted to *Geoscientific Model Development*.

Your consideration is very thorough. We have cited two papers following the statement, and the corresponding references have been added. Besides, acknowledgements to you and the three anonymous referees for your valuable comments have been included in the revised manuscript. Please find our detailed point-by-point responses to the comments below. The editor's comments are in black, and our responses are in red. A tracked-changes version of the revised manuscript is also uploaded for your reference.

Thank you again for spending your time in handling this manuscript.

Best regards,

Yixiong Lu and all co-authors
* * *
**Response to Topical Editor**

Comments to the Author:

Dear Authors,

Thank you for your comments and your revised version, which respond to the reviewer question. Your paper is now almost ready for publication.

I still have a little concern. Line 492, you state "The double ITCZ is a prominent bias in CGCM, which may result from discrepancies in the representation of both atmospheric and oceanic processes". I strongly agree, but I'm not sure that this a widely know. Some references about that could be useful. Maybe the reference in the following sentence are relevant, but I didn't check.

Kind regards, and than you for publishing in GMD.

Olivier Marti
GMD Topical Editor

Thank you for the thorough consideration. We have cited two papers following the statement, as follows,

"The double ITCZ is a prominent bias in CGCM, which may result from discrepancies in the representation of both atmospheric and oceanic processes (Zhang et al., 2019; Song and Zhang, 2020)."

The corresponding two references are also included, one of which is newly added as follows,

Song, X. and Zhang, G. J.: Role of equatorial cold tongue in central Pacific double-ITCZ bias in the NCAR CESM1.2, J. Climate, 33, 10407-10418, doi:10.1175/JCLI-D-20-0141.1, 2020.

In addition, we thank the editor and the three anonymous referees for your valuable comments that helped to improve the manuscript. We have included these words in the section of Acknowledgements.

[revised manuscript text omitted]